# SCALING LAWS FOR DIFFUSION TRANSFORMERS

## ABSTRACT

Diffusion transformers (DiT) have already achieved appealing synthesis and scaling properties in content recreation, *e.g.,* image and video generation. However, scaling laws of DiT are less explored, which usually offer precise predictions regarding optimal model size and data requirements given a specific compute budget. Therefore, experiments across a broad range of compute budgets, from `1e17` to `6e18` FLOPs are conducted to confirm the existence of scaling laws in DiT *for the first time*. Concretely, the loss of pretraining DiT also follows a power-law relationship with the involved compute. Based on the scaling law, we can not only determine the optimal model size and required data but also accurately predict the text-to-image generation loss given a model with 1B parameters and a compute budget of `1e21` FLOPs. Additionally, we also demonstrate that the trend of pretraining loss matches the generation performances (*e.g.,* FID), even across various datasets, which complements the mapping from compute to synthesis quality and thus provides a predictable benchmark that assesses model performance and data quality at a reduced cost.

## 1 INTRODUCTION

Scaling laws in large language models (LLMs) (Kaplan et al., 2020; Hestness et al., 2017; Henighan et al., 2020; Hoffmann et al., 2022) have been widely observed and validated, suggesting that pretraining performance follows a power-law relationship with the compute $C$. The actual compute could be roughly calculated as $C = 6ND$, where $N$ is the model size and $D$ is the data quantity. Therefore, determining the scaling law helps us make informed decisions about resource allocation to maximize computational efficiency, namely, figure out the optimal balance between model size and training data (*i.e.,* the optimal model and data scale) given a fixed compute budget. However, scaling laws in diffusion models remain less explored.

The scalability has already been demonstrated in diffusion models, especially for diffusion transformers (DiT). Specifically, several prior works (Mei et al., 2024; Li et al., 2024) reveal that larger models always result in better visual quality and improved text-image alignment. However, the scaling property of diffusion transformers is *clearly observed but not accurately predicted*. Besides, the absence of explicit scaling laws also hinders a comprehensive understanding of how training budget relate to model size, data quantity, and loss. As a result, we cannot determine accordingly the optimal model and data sizes for a given compute budget and accurately predict training loss. Instead, heuristic configuration searches of models and data are required, which are costly and challenging to ensure optimal balance.

In this work, we characterize the scaling behavior of diffusion models for text-to-image synthesis, resulting in the explicit scaling laws of DiT *for the first time*. To investigate the explicit relationship between pretraining loss and compute, a wide range of compute budgets from `1e17` to `6e18` FLOPs are used. Models ranging from 1M to 1B are pretrained under given compute budgets. As shown in Fig. 1, for each compute budget, we can fit a parabola and extract an optimal point that corresponds to the optimal model size and consumed data under that specific compute constraint. Using these optimal configurations, we derive scaling laws by fitting a power-law relationship between compute budgets, model size, consumed data, and training loss. To evaluate the derived scaling laws, we extrapolate the compute budget to `1.5e21` FLOPs that results in the compute-optimal model size (approximately 1B parameters) and the corresponding data size. Therefore, a 1B-parameter model is trained under this budget and the final loss matches our prediction, demonstrating the effectiveness and accuracy of our scaling laws.

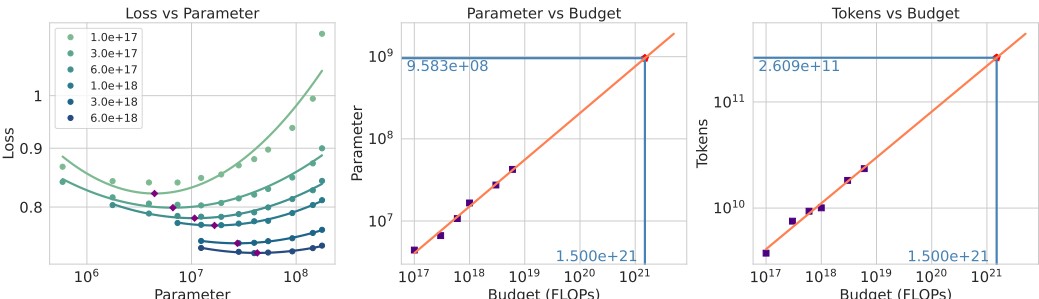

Figure 1: **IsoFLOP and Model/Data Scaling Curves.** For each training budget, we train multiple models of varying sizes. A parabola is fitted to the loss for each training budget, and the minimum point on each parabola (represented by the purple dots) corresponds to the optimal allocation of model size and data for that specific budget. By identifying the model and data sizes at these optimal points, we can plot the scaling trends of model parameters, tokens, and training budgets. The power-law curves shown allow us to predict the optimal configurations for larger compute budgets, such as `1.5e21` FLOPs.

To make the best use of the scaling laws, we demonstrate that the generation performances (*e.g.,* FID (Fréchet Inception Distance)) also match the trend of pretraining loss. Namely, the synthesis quality also follows the power-law relationship with the compute budget, making it predictable. More importantly, this observation is transferable across various datasets. We conduct additional experiments on the COCO validation set (Lin et al., 2015), and the same scaling patterns hold, even when tested on out-of-domain data. Accordingly, scaling laws could serve as a predictable benchmark where we can assess the quality of both models and datasets at a significantly reduced computational cost, enabling efficient evaluation and optimization of the model training process.

To summarize, we at first confirm the presence of scaling laws in diffusion transformers during training, revealing a clear power-law relationship between compute budget and training losses. Next, we establish a connection between pretraining loss and synthesis evaluation metrics. Finally, We conducted preliminary experiments that demonstrate the potential of using scaling laws to evaluate both data and model performance. By conducting scaling experiments at a relatively low cost, we can assess and validate the effectiveness of different configurations based on the fitted power-law coefficients.

## 2 RELATED WORK

**Diffusion Transformers**    Transformers have become the de facto model architecture in language modeling (Radford, 2018; Radford et al., 2019; Devlin et al., 2019), and they have also achieved significant success in computer vision tasks (Dosovitskiy et al., 2021; He et al., 2021). Recently, Transformers have been introduced into diffusion models (Peebles & Xie, 2023), where images are divided into patches (tokens), and the diffusion process is learned on these tokens. Additional conditions, such as timestep and text, are incorporated into the network via cross-attention (Chen et al., 2023), Adaptive Normalization (Perez et al., 2017), or by concatenating them with image tokens (Bao et al., 2023). Zheng et al. (2023) proposed masked transformers to reduce training costs, while Lu et al. (2024) introduced techniques for unrestricted resolution generation. Diffusion Transformers (DiTs) inherit the scalability, efficiency, and high capacity of Transformer architectures, positioning them as a promising backbone for diffusion models. Motivated by this scalability, we investigate the scaling laws governing these models in this work. To ensure robust and clear conclusions, we adopt a vanilla Transformer design (Vaswani et al., 2023), using a concatenation of image, text, and time tokens as input to the models.

**Scaling Laws**    Scaling laws (Hestness et al., 2017) have been fundamental in understanding the performance of neural networks as they scale in size and data. This concept has been validated across several large pretraining models (Dubey et al., 2024; Bi et al., 2024; Achiam et al., 2023). Kaplan et al. (2020); Henighan et al. (2020) were the first to formalize scaling laws in language

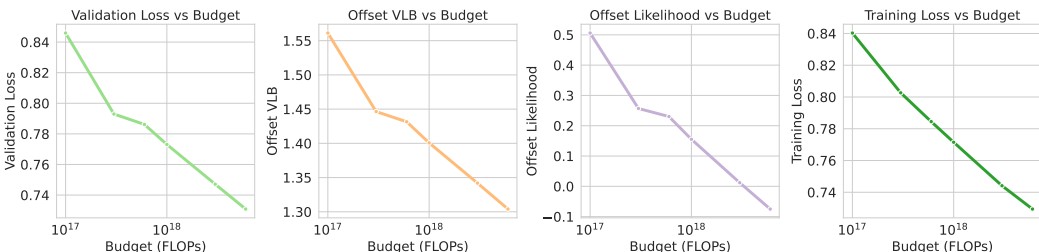

Figure 2: **Scaling Curves for Different Metrics.** We present the scaling curves for training and validation loss, offset VLB, and offset likelihood. Validation metrics are evaluated on the COCO 2014 validation set. The metrics display consistent trends and similar shapes, all adhering to a power-law relationship. This demonstrates that each of these metrics can be used to observe scaling laws effectively. For simplicity, we primarily focus on training loss in subsequent analyses.

models and extend them to autoregressive generative models, demonstrating that model performance scales predictably with increases in model size and dataset quantity. Hoffmann et al. (2022) further highlighted the importance of balancing model size and dataset size to achieve optimal performance. In the context of diffusion models, prior works (Mei et al., 2024; Li et al., 2024) have empirically demonstrated their scaling properties, showing that larger compute budgets generally result in better models. These studies also compared the scaling behavior of various model architectures and explored sampling efficiency. However, no previous works provide an explicit formulation of scaling laws for diffusion transformers to capture the relationship between compute budget, model size, data, and loss. In this paper, we aim to address this gap by systematically investigating the scaling behavior of diffusion transformers (DiTs), offering a more comprehensive understanding of their scaling properties.

## 3 METHOD

### 3.1 BASIC SETTINGS

Unless otherwise stated, all experiments in this paper adhere to the same basic settings. While the training techniques and strategies employed may not be optimal, they primarily affect the scaling coefficients rather than the scaling trends. In this section, we outline the critical settings used in our experiments. Additional details are provided in Appendix C.

**Diffusion Formulation**    All experiments are conducted using the Rectified Flow (RF) formulation (Liu et al., 2022; Lipman et al., 2022; Albergo et al., 2023) with **v**-prediction. For timestep sampling, we adopt the Logit-Normal (LN) Sampling scheduler $\pi_{ln}(t; m, s)$, as proposed in Esser et al. (2024). A detailed ablation study of this choice can be found in Appendix F.1.

**Models & Dataset**    As noted in Kaplan et al. (2020), the model design does not significantly impact scaling trends unless the architecture is extremely shallow or slim. We employ a vanilla transformer architecture (Vaswani et al., 2023) with minimal modifications. The text, image, and timestep tokens are concatenated as input (in-context conditioning (Peebles & Xie, 2023)). A dataset of 108 million image-text pairs is randomly sampled from Laion-Aesthetic (Schuhmann et al., 2022). This subset is then re-captioned using LLAVA 1.5 (Liu et al., 2024). A validation set consisting of 1 million pairs is created from this subset. Further details can be found in Appendix C.1 and C.2.

**Scaling Metrics**    A natural question arises when investigating scaling laws during training: *What metrics should be selected to observe scaling behavior?* In the context of Large Language Models (LLMs), the standard approach is autoregressive training (Radford, 2018; Radford et al., 2019), where the model is trained to predict the next token in a sequence, directly optimizing the likelihood of the training data. This has proven to be a reliable method for measuring model performance as the compute budget scales up. Inspired by this approach, we extend the concept of scaling laws to diffusion models, using **loss** and **likelihood** as our key metrics.

**Loss** is the primary metric chosen to observe scaling behavior during training. Unlike autoregressive models, diffusion models do not directly optimize likelihood. Instead, the objective is to match a time-conditioned velocity field (Ma et al., 2024; Liu et al., 2022; Lipman et al., 2022; Albergo et al., 2023). Specifically, the velocity $\mathbf{v}(x_t, t)$ at timestep $t$ is defined as:

$$\mathbf{v}(x_t, t) = x_t' = \alpha_t' x_0 + \beta_t' \epsilon, \tag{1}$$

where $x_0$ represents the original data and $\epsilon$ denotes the noise. Here, the prime symbol $'$ indicates the derivative with respect to time $t$. In the rectified flow framework, the coefficients $\alpha_t$ and $\beta_t$ are defined as $\alpha_t = 1 - t, \beta_t = t$. Thus, the velocity $\mathbf{v}$ can be further simplified as:

$$\mathbf{v}(x_t, t) = -x_0 + \epsilon. \tag{2}$$

The corresponding loss function is expressed in terms of the expected value:

$$\mathcal{L}(\theta, \mathbf{x}, t) = \mathbb{E}_{t \sim [1,T], \epsilon \sim \mathcal{N}(0,I)} \left[ \|\mathbf{v}_\theta(x_t, t) + x_0 - \epsilon\|^2 \right] \approx \frac{1}{N} \sum_{i=1}^{N} \left[ \|\mathbf{v}_\theta(x_t, t) + x_0 - \epsilon\|^2 \right]. \tag{3}$$

The **training loss** is estimated using a Monte Carlo method, which involves timesteps and noise sampling. The stochasticity inherent in this process can cause significant fluctuations, which are mitigated by employing a larger batch size of 1024 and applying Exponential Moving Average (EMA) smoothing. In our experiments, we set $\alpha_{\text{EMA}} = 0.9$, which is found to produce stable results. A detailed ablation study on the choice of EMA coefficients is provided in Appendix F.2. This smoothing procedure helps reduce variance and provides clearer insights into training dynamics.

In addition to the training loss, **validation loss** is also computed on the COCO 2014 dataset (Lin et al., 2015). To ensure consistency with the training loss, timesteps are sampled using the LN timestep sampler $\pi_{ln}(t; m, s)$, and evaluation is performed on 10,000 data points, with 1,000 timesteps sampled per point.

**Likelihood** is our secondary metric. The likelihood over the dataset distribution $P_\mathcal{D}$ given model parameters $\theta$ is represented as $\mathbb{E}_{x \sim P_\mathcal{D}}[p_\theta(x)]$, which can be challenging to compute directly. In this paper, we measure likelihood using two different methods. The first method is based on the Variational Autoencoder (VAE) framework (Kingma et al., 2021; Song et al., 2021a; Vahdat et al., 2021), which approximates the lower bound of log-likelihood using the Variational Lower Bound (VLB). Since the VAE component in our experiments is fixed to Stable Diffusion 1.5, terms related to the VAE remain constant and are ignored in our computation, as they do not affect the scaling behavior. The second method uses Neural Ordinary Differential Equations (ODEs) (Chen et al., 2019; Grathwohl et al., 2018), enabling the computation of exact likelihood through reverse-time sampling.

**Variational Lower Bound (VLB)** We estimate the VLB following the approach in Kingma et al. (2021), where it is treated as a reweighted version of the validation loss. The process starts by converting the estimated velocity into a corresponding estimate of $x_0$, after which the loss is computed based on the difference between the estimated $x_0$ and the original sample. To obtain the VLB, this loss is further reweighted with the weighting coefficient being the derivative of signal-to-noise ratio of noisy samples with respect to time $t$, *i.e.,* $\text{SNR}'(t)$. More details can be found in Appendix D. All models are evaluated on the COCO 2014 validation set using 10,000 data pairs. For each data point, 1,000 timesteps are sampled to ensure accurate estimations of the VLB.

**Exact Likelihood** The exact likelihood is computed using reverse-time sampling, where a clean sample is transformed into Gaussian noise. The accumulated likelihood transition is calculated using the instantaneous change of variables theorem (Chen et al., 2019):

$$\log p_\theta(x) = \log p_\theta(x_T) - \int_0^T \nabla \cdot f_\theta(x_t, t) \, dt,$$

where $f_\theta(x_t, t)$ represents the model's output, and $\log p_\theta(x_T)$ is the log density of the final Gaussian noise. The reverse process evolves $t$ from 0 (data points) to $T$ (noise). The reverse sampling is performed over 500 steps using the Euler method.

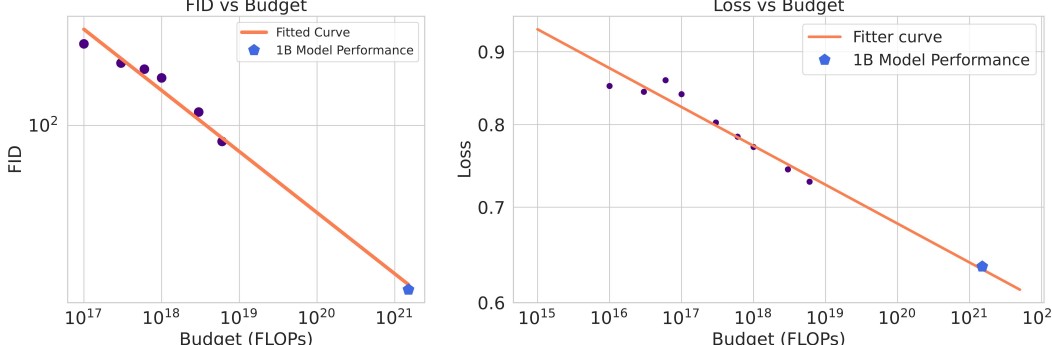

Figure 3: **Performance Scaling Curves.** The plot shows the relationship between training loss, FID, and compute budget, with both axes displayed on a logarithmic scale. The orange lines represent the fitted power-law curves for both metrics across various budgets, while the purple dots mark the performance of compute-optimal models at smaller budgets. In both cases, the blue pentagons indicate the predicted performance at a budget of `1.5e21` FLOPs, demonstrating highly accurate predictions for large-scale models.

Following model training, we evaluate models on the validation set to assess their compute-optimal performance. As illustrated in Fig. 2, all four metrics (training loss, validation loss, offset VLB, and offset exact likelihood ) exhibit similar trends and shapes as the training budget increases, showing their utility in observing scaling laws. These consistent patterns suggest that any of these metrics can be effectively used to monitor scaling behavior. However, to simplify our experimental workflow, we prioritize **training loss** as the primary metric. Training loss can be observed directly during the training process, without the need for additional evaluation steps, making it a more practical choice for tracking scaling laws in real-time.

### 3.2 SCALING LAWS IN TRAINING DIFFUSION TRANSFORMERS

**Scaling Laws** In this section, we investigate the scaling laws governing diffusion transformers, which describe the relationships between several key quantities: the objective function, model parameters, tokens, and compute. The objective measures discrepancy between the data and model's predictions. The number of parameters $N$ reflects model's capacity, while tokens $D$ denote the total amount of data (in tokens) processed during training. Compute $C$, typically measured in Floating Point Operations (FLOPs), quantifies the total computational resources consumed. In our experiments, the relationship between compute, tokens, and model size is formalized as $C = 6ND$, directly linking the number of parameters and tokens to the overall compute budget.

Building on this, we hypothesize that power law equations can effectively capture the scaling relationships between these quantities. Specifically, we represent the optimal model size and token count as functions of the compute budget as follows:

$$N_{\text{opt}} \propto C^a \quad \text{and} \quad D_{\text{opt}} \propto C^b, \tag{4}$$

where $N_{\text{opt}}$ and $D_{\text{opt}}$ denote the optimal number of parameters and tokens for a given compute budget $C$, with $a$ and $b$ as scaling exponents that describe how these quantities grow with compute.

To empirically verify these scaling relationships, following Approach 2 in Hoffmann et al. (2022), we plot the isoFLOP figure to explore scaling laws. We select compute budgets [`1e17`, `3e17`, `6e17`, `1e18`, `3e18`, `6e18`]. We change the In-context Transformers from 2 layers to 15 layers. For all experiments, we use AdamW (Kingma & Ba, 2017; Loshchilov & Hutter, 2019) as the default optimizer. Except for the 1B parameter prediction experiments, we apply a constant learning rate of 1e-4. In the 1B parameter experiments, we train the model with a learning rate of `1e-4` for 80% of the total iterations, then decay it to `3.16e-5` for the next 10%, and finally to `1e-5` for the last 10%. This learning rate decay is crucial for larger models, leading to more accurate predictions. Following Wortsman et al. (2023); Team (2024); Molybog et al. (2023), we use a weight decay of 0.01, an epsilon value of `1e-15`, and betas [0.9, 0.95]. For all experiments, we employ a batch size

of 1024 and apply gradient clipping with a threshold of 1.0. To enable classifier-free guidance (Ho & Salimans, 2022), we randomly drop the label with a probability of 0.1. During training, we use mixed precision with the BF16 data format.

For each budget, we fit a parabola to the resulting performance curve, as illustrated in Fig. 1, to identify the optimal loss, model size, and data allocation (highlighted by the purple dots). By collecting the optimal values from different budgets and fitting them to a power law, we establish relationships between the optimal loss, model size, data, and compute budgets.

As shown in Fig. 1, except for the `1e17` budget, the parabolic fits align closely with the empirical results. This analysis confirms that the optimal number of parameters and tokens scale with the compute budget according to the following expressions:

$$N_{\text{opt}} = 0.0009 \cdot C^{0.5681}, \tag{5}$$

$$D_{\text{opt}} = 186.8535 \cdot C^{0.4319}. \tag{6}$$

In this way, we establish the relationship between compute budget and model/data size. And from the fitted scaling curves, we observe that the ratio between the scaling exponent for data and the scaling exponent for model size is $0.4319/0.5681$. This indicates that, under our specific settings, both the model and data sizes need to increase in tandem as the training budget increases. However, the model size grows at a slightly faster rate compared to the data size, as reflected by the proportional relationship between the two exponents.

Additionally, in Fig. 3, we fit the relationship between the compute budget and loss, which follows the equation:

$$L = 2.3943 \cdot C^{-0.0273}. \tag{7}$$

To validate the accuracy of these fitted curves, we calculate the optimal model size (958.3M parameters) and token count for a compute budget of `1.5e21`. A model is then trained with these specifications to compare its training loss with the predicted value. As demonstrated in Fig. 3, this model's training loss closely matches the predicted loss, further confirming the validity of our scaling laws.

### 3.3 Predicting Generation Performance

In generative tasks, relying solely on training loss to evaluate performance often falls short of capturing the perceptual quality of generated images. Training loss does not fully reflect how closely the generated images resemble real ones or how well they align with textual prompts in terms of content and style. To address this limitation, the Fréchet Inception Distance (FID) is commonly used to evaluate image quality.

**Fréchet Inception Distance (FID)**   FID measures the distance between the distributions of generated and real images in a feature space, with lower FID values indicating higher image quality. For our experiments, we utilize compute-optimal models at each training budget, generating samples using the Euler discretization method with 25 steps and a classifier-free guidance (CFG) scale of 10.0. Details on the impact of steps and CFG scale are provided in Appendix F.3. To compute FID, we follow the approach of Sauer et al. (2021), using CLIP features instead of traditional Inception features. Specifically, we employ ViT-L/14 Dosovitskiy et al. (2021) as the vision encoder to extract features from both generated and dataset images and compare their statistics.

**Scaling laws for Performance Predictions**   Our analysis reveals that the relationship between FID and the training budget follows a clear power-law trend, as shown in Fig. 3 (left). The relationship is captured by the following equation:

$$\texttt{FID} = 2.2566 \times 10^6 \cdot C^{-0.234}, \tag{8}$$

where $C$ is the training budget. The purple dots in the figure represent the FID scores of compute-optimal models at various budgets, and the orange line represents the fitted power-law curve. Notably, the prediction for FID at a large budget of `1.5e21` FLOPs (blue pentagon) is highly accurate, confirming the reliability of scaling laws in forecasting model performance even at larger scales. As the compute budget increases, FID values decrease consistently, demonstrating that scaling laws can effectively model and predict the quality of generated images as resources grow.

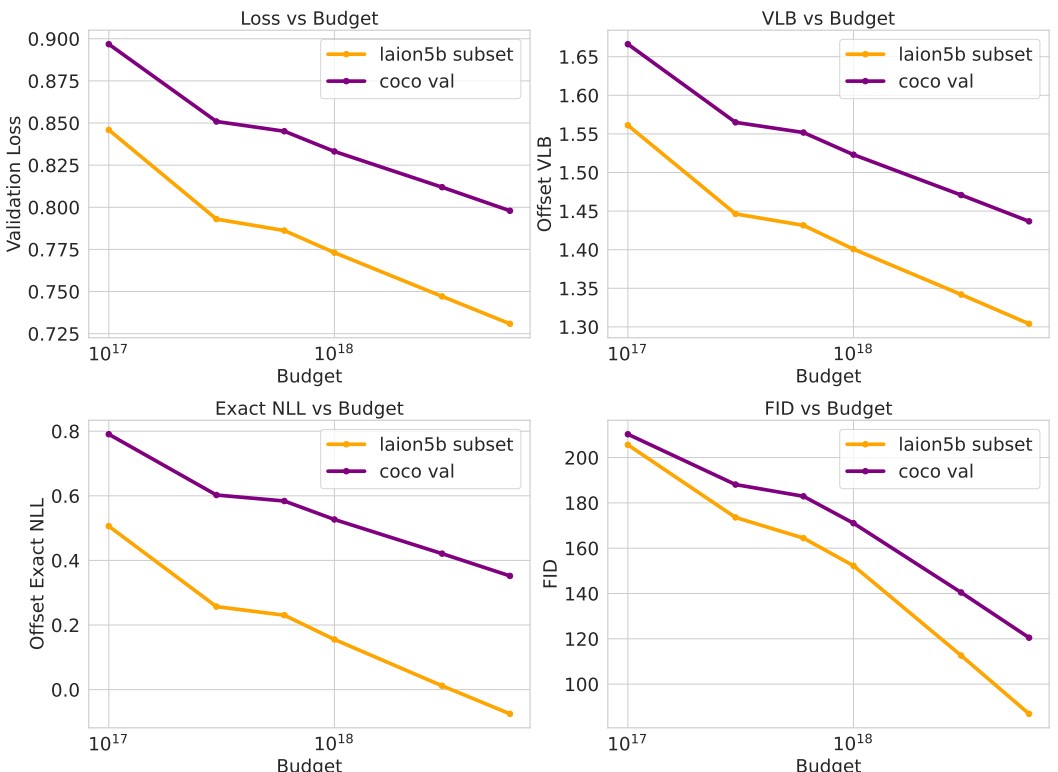

Figure 4: **Scaling laws for OOD data**. The evaluation of loss, VLB, and NLL are conducted on an out-of-domain dataset COCO 2014 validation set. All metrics decrease monotonically as the budget increases and they share a similar shape. However, a notable shift can be observed. Models have worse performance on the COCO validation set since they are trained on a different data distribution. The shift reflects the differences between datasets.

## 3.4 SCALING LAWS FOR OUT-OF-DOMAIN DATA

Scaling laws remain valid even when models are tested on out-of-domain datasets. To demonstrate this, we conduct validation experiments on the COCO 2014 validation set (Lin et al., 2015), using models that were trained on the Laion5B subset. In these experiments, we evaluate four key metrics: validation loss, Variational Lower Bound (VLB), exact likelihood, and Fréchet Inception Distance (FID). Each metric is tested on 10,000 data points to examine the transferability of the scaling laws across datasets.

The results, as shown in Fig. 4, reveal two key observations:

- **Consistent Trends**: Across all four metrics (validation loss, VLB, exact likelihood, and FID), the trends are consistent between the Laion5B subset and the COCO validation dataset. As the training budget increases, model performance improves in both cases, indicating that scaling laws generalize effectively across datasets, regardless of domain differences.

- **Vertical Offset**: There is a clear vertical offset between the metrics for the Laion5B subset and the COCO validation dataset, with consistently higher metric values observed on the COCO dataset. This suggests that while scaling laws hold, the absolute performance is influenced by dataset-specific characteristics, such as complexity or distribution. For metrics like validation loss, VLB, and exact likelihood, this offset remains relatively constant across different training budgets. The gap between the FID values for the Laion5B subset and the COCO validation set widens as the training budget increases. However, the relationship between FID and budget on the COCO validation set still follows a power-law

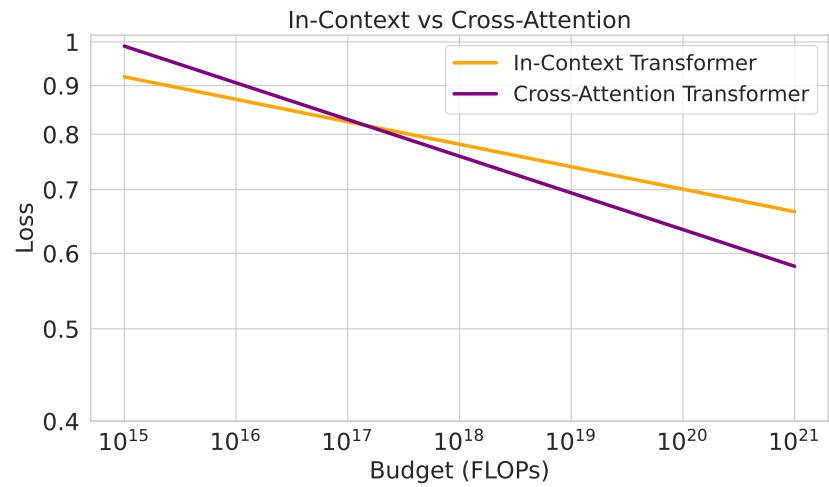

Figure 5: **Scaling curves for In-Context and Cross-Attention Transformers.** The plot compares the scaling behavior of In-Context Transformers and Cross-Attention Transformers with respect to compute budget (FLOPs). In-context transformers, which concatenate image, text, and time tokens, show a more gradual decline in loss compared to Cross-Attention Transformers, which inject time and text tokens via AdaLN and cross-attention blocks. The steeper slope of Cross-Attention Transformers indicates more efficient performance improvement with increasing compute, meaning they achieve lower loss with the same budget. These results highlight the efficiency of Cross-Attention Transformers and illustrate the potential of scaling laws in predicting performance trends across different model architectures.

> trend. This suggests that, even on out-of-domain data, the FID-budget relationship can be reliably modeled using a power law, allowing us to predict the model's FID for a given budget.

In summary, these results demonstrate that scaling laws are robust and can be applied effectively to out-of-domain datasets, maintaining consistent trends while accounting for dataset-specific performance differences. Despite the vertical offset in absolute performance, particularly in metrics like FID, the power-law relationships remain intact, allowing for reliable predictions of model performance under varying budgets. These findings highlight the potential of scaling laws as a versatile tool for understanding model behavior across datasets. The ability to project model efficiency and performance onto new data domains underscores the broader applicability of scaling laws in real-world scenarios.

## 4 SCALING LAWS AS A PREDICTABLE BENCHMARK

Scaling laws offer a robust framework for evaluating the design quality of both models and datasets. Previous work such as Dubey et al. (2024); Bi et al. (2024) have all explored the scaling laws in data mix and quality. By modifying either the model architecture or the data pipeline, isoFLOP curves can be generated at smaller compute budgets to assess the impact of these changes. Specifically, after making adjustments to the model or dataset, experiments can be conducted across a range of smaller compute budgets, and the relationship between compute and metrics such as loss, parameter count, or token count can be fitted. The effectiveness of these modifications can then be evaluated by analyzing the exponents derived from the power-law fits.

Our evaluation follows three key principles:

- **Model improvements**: With a fixed dataset, a more efficient model will exhibit a lower model scaling exponent and a higher data scaling exponent. This suggests that the model can more effectively utilize the available data, allowing for a greater focus on increasing the dataset size with additional compute resources.

- **Data improvements**: When the model is fixed, a higher-quality dataset will result in a lower data scaling exponent and a higher model scaling exponent. This implies that a better dataset enables the model to scale more efficiently, yielding superior results with fewer resources.

- **Loss improvements**: Across both model and data modifications, an improved training pipeline is reflected in a smaller loss scaling exponent relative to compute. This indicates that the model achieves better performance with less compute, signaling overall gains in training efficiency.

The scaling trends for both models, In-Context Transformers and Cross-Attention Transformers, are clearly illustrated in Fig. 5. The loss scaling curves demonstrate that as the compute budget increases, the performance of both models improves, but at different rates. The Cross-Attention Transformer shows a steeper decline in loss compared to the In-Context Transformer, indicating that it achieves better performance with the same amount of compute.

The fitted scaling curves, as summarized in Tab. 1, support this observation. The Cross-Attention Transformer exhibits a larger model exponent, meaning that as compute budgets increase, more resources should be allocated toward scaling the dataset. Additionally, the smaller loss exponent of the Cross-Attention Transformer suggests a more rapid decline in loss, indicating that this model achieves superior performance compared to the In-Context Transformer. These findings align with the conclusions of Peebles & Xie (2023).

| Model | Model Exponent | Data Exponent | Loss Exponent |
|---|---|---|---|
| In-context | 0.56 | 0.43 | -0.0273 |
| Cross-Attention | 0.54 | 0.46 | -0.0385 |

Table 1: Exponents of model, data, and loss for different model architectures.

This example illustrates how scaling laws can serve as a reliable and predictable benchmark for evaluating the effectiveness of both model architectures and datasets. By analyzing the scaling exponents, we can draw meaningful conclusions about the potential and efficiency of different design choices in model and data pipelines.

## 5 DISCUSSION

**Limitations**   While our work demonstrates the presence of scaling laws in Diffusion Transformers (DiT), several limitations must be acknowledged. First, we used fixed hyperparameters, such as learning rate and batch size, across all experiments. A more comprehensive investigation into how these parameters should be adjusted as budgets scale may lead to more precise predictions. Second, our study focuses solely on text-to-image generation, leaving the scalability of Diffusion Transformers with other data modalities, such as video, unexplored. Finally, we limited our analysis to In-Context Transformers and Cross-Attention Transformers, leaving the evaluation of additional model variants to future research.

**Conclusion**   In this work, we explored the scaling laws of Diffusion Transformers (DiT) across a broad range of compute budgets, from `1e17` to `6e18` FLOPs, and confirmed the existence of a power-law relationship between pretraining loss and compute. This relationship enables precise predictions of optimal model size, data requirements, and model performance, even for large-scale budgets such as `1e21` FLOPs. Furthermore, we demonstrated the robustness of these scaling laws across different datasets, illustrating their generalizability beyond specific data distributions. In terms of generation performance, we showed that training budgets can be used to predict the visual quality of generated images, as measured by metrics like FID. Additionally, by testing both In-context Transformers and Cross-Attention Transformers, we validated the potential of scaling laws to serve as a predictable benchmark for evaluating and optimizing both model and data design, providing valuable guidance for future developments in text-to-image generation using DiT.

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
