APPENDIX

This appendix is organized as follows:

- In Section A, we list an overview of the notation used in the paper.

- In Section B, we provide extended related work.

- In Section C, we provide experimental details.

- In Section D, we provide the derivation of the likelihood used in this paper.

- In Section E, we provide the details of scaling FLOPs counting based on our models.

- In Section F, we demonstrate ablative studies during the experiments.

- In Section G, we add some more results to respond the reviewers' questions.

# A NOTATION

Tab. 2 provides an overview of the notation used in this paper.

| Symbol | Meaning |
|--------|---------|
| RF | Rectified Flow (Lipman et al., 2022; Liu et al., 2022; Albergo et al., 2023) |
| LN | Logit-Normal timestep sampler $\pi_{ln}(t; m, s)$ (Esser et al., 2024) |
| SNR | Signal-Noise-Ratio, $\lambda_t = \frac{\alpha_t^2}{\beta_t^2}$ |
| DDPM | Denoising Diffusion Probabilistic Models (Ho et al., 2020) |
| LDM | Latent Diffusion Models (Rombach et al., 2022) |
| VP | Variance Preserving formulation (Song et al., 2021b) |
| VLB | Variational Lower Bound |
| VAE | Variational Autoencoder |
| KL | KL Divergence $KL(p(x)|q(x)) = \mathbb{E}_{p(x)}[ln\frac{p(x)}{q(x)}]$ |
| $P_{\mathcal{D}}$ | Dataset distribution |
| $d_{attn}$ | Dimension of the attention output |
| $d_{ff}$ | Dimension of the Feedforward layer |
| $d_{model}$ | Dimension of the residual stream |
| $n_{layer}$ | Depth of the Transformer |
| $l_{ctx}$ | Context length of input tokens |
| $l_{img}$ | Context length of image tokens |
| $l_{text}$ | Context length of text tokens |
| $l_{time}$ | Context length of time tokens |
| $N_{voc}$ | Size of Vocabulary list |
| $n_{head}$ | Number of heads in Multi-Head Attention |
| $N$ | Number of parameters |
| $D$ | Size of training data (token number) |
| $C$ | Compute budget, $C = 6ND$ |
| $N_{opt}$ | Optimal number of parameters for the given budget |
| $D_{opt}$ | Optimal training tokens for the given budget. |
| $\boldsymbol{\epsilon}$ | Gaussian Noise $\mathcal{N}(0, I)$ |
| $\mathbf{x}_t$ | A sample created at timestep $t$ |
| t | Timestep ranging from [0, 1] |
| $\mathbf{v}$ | Velocity $\mathbf{v} = \mathbf{x}_1 - \mathbf{x}_0$ |
| $\alpha_t$ | $\alpha_t$ represents the scaling factor during noise sample creation. |
| $\beta_t$ | $\beta_t$ represents the diffusion factor during noise sample creation. |
| $\sigma_t$ | Noise Level defined for each timestep |
| $f_{\boldsymbol{\theta}}(\mathbf{x})$ | The network we use to learn the transition kernel. $f_{\boldsymbol{\theta}} : \mathbb{R}^{N \times M} \to \mathbb{R}^{N \times M}$ |
| $\eta$ | Learning rate |
| $\pi_{ln}(t; m, s)$ | Logit-Normal Timestep sampling schedule, $m$ is the location parameter, $s$ is the scale parameter |
| $\mathcal{L}(\boldsymbol{\theta}, \mathbf{x}, t)$ | Loss given model parameters, data points, and timesteps. |
| $\lambda$ | Fixed step size for ODE/SDE samplers |
| $\boldsymbol{\theta}$ | Parameters for diffusion models |
| $\phi$ | Parameters for VAE encoder |
| $\psi$ | Parameters for VAE decoder |
| $\alpha_{\mathrm{EMA}}$ | EMA coefficient |

Table 2: **Summary of the notation and abbreviations used in this paper.**

## B  EXTENDED RELATED WORK

**Diffusion Models**  Diffusion models have gained significant attention due to their effectiveness in generative modeling, starting from discrete-time models (Sohl-Dickstein et al., 2015; Ho et al., 2020; Song & Ermon, 2020) to more recent continuous-time extensions (Song et al., 2021b). The core idea of diffusion models is to learn a sequence of noise-adding and denoising steps. In the forward process, noise is gradually added to the data, pushing it toward a Gaussian distribution, and in the reverse process, the model learns to iteratively denoise, recovering samples from the noise. Continuous-time variants (Song et al., 2021b) further generalize this framework using stochastic differential equations (SDEs), allowing for smoother control over the diffusion process. These methods leverage neural network architectures to model the score function and offer flexibility and better convergence properties compared to discrete versions. Diffusion models have shown remarkable success in various applications. For instance, ADM (Dhariwal & Nichol, 2021) outperforms GAN on ImageNet. Following this success, diffusion models have been extended to more complex tasks such as text-to-image generation. Notably, models like Stable Diffusion (Rombach et al., 2022) and DALLE (Ramesh et al., 2021) have demonstrated the ability to generate highly realistic and creative images from textual descriptions, representing a significant leap in the capabilities of generative models across various domains.

**Normalizing Flows**  Normalizing flows has been a popular generative modeling approach due to their ability to compute exact likelihoods while providing flexible and invertible transformations. Early works like GLOW (Kingma & Dhariwal, 2018) and RealNVP (Dinh et al., 2017) introduced powerful architectures that allowed for efficient sampling and likelihood estimation. However, these models were limited by the necessity of designing specific bijective transformations, which constrained their expressiveness. To address these limitations, Neural ODE (Chen et al., 2019) and FFJORD (Grathwohl et al., 2018) extended normalizing flows to the continuous domain using differential equations. These continuous normalizing flows (CNFs) allowed for more flexible transformations by parameterizing them through neural networks and solving ODEs. By modeling the evolution of the probability density continuously, these methods achieved a higher level of expressiveness and adaptability compared to their discrete counterparts. Recent work has begun to bridge the gap between continuous normalizing flows and diffusion models. For instance, ScoreSDE (Song et al., 2021b) demonstrated how the connection between diffusion models and neural ODEs can be leveraged, allowing both exact likelihood computation and flexible generative processes. More recent models like Flow Matching (Lipman et al., 2022) and Rectified Flow (Liu et al., 2022) further combined the strengths of diffusion and flow-based models, enabling efficient training via diffusion processes while maintaining the ability to compute exact likelihoods for generated samples. In this paper, we build upon the formulation introduced by rectified flow and Flow Matching. By leveraging the training approach of diffusion models, we benefit from their generative performance, while retaining the capability to compute likelihoods.

**Likelihood Estimation**  Likelihood estimation in diffusion models can be approached from two primary perspectives: treating diffusion models as variational autoencoders (VAEs) or as normalizing flows. From the VAE perspective, diffusion models can be interpreted as models where we aim to optimize a variational lower bound (VLB) on the data likelihood (Kingma & Welling, 2022). The variational lower bound decomposes the data likelihood into a reconstruction term and a regularization term, where the latter measures the divergence between the approximate posterior and the prior. In diffusion models, this framework allows us to approximate the true posterior using a series of gradually noised latent variables. Recent works (Ho et al., 2020; Kingma et al., 2021; Song et al., 2021a) have derived tighter bounds for diffusion models, enabling more accurate likelihood estimation by optimizing this variational objective. Alternatively, diffusion models can be viewed as a form of normalizing flows, particularly in the context of continuous-time formulations. Using neural ODEs (Chen et al., 2019), diffusion models can be trained to learn exact likelihoods by modeling the continuous reverse process as an ODE. By solving this reverse-time differential equation, one can directly compute the change in log-likelihood through the flow of probability densities (Grathwohl et al., 2018). This approach provides a method for exact likelihood computation, bridging the gap between diffusion models and normalizing flows, and offering a more precise estimate of the likelihood for generative modeling.

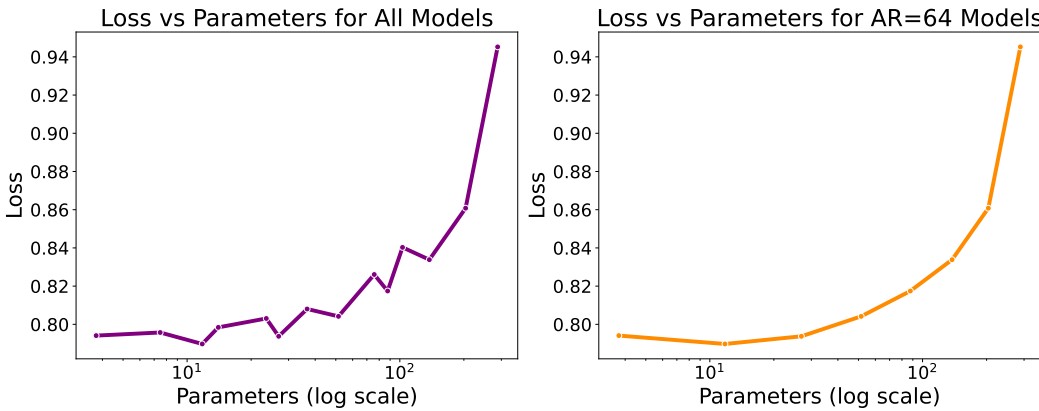

Figure 6: The effect of AR.

## C   EXPERIMENTAL DETAILS

### C.1   DATA

We primarily utilized three datasets in our work. Several ablation studies on formulation and model design were conducted using JourneyDB (Sun et al., 2023). Additionally, we curated a subset of 108 million image-text pairs from the Laion-Aesthetic dataset, applying a threshold of 5 for the aesthetic score. After collecting the data, we re-captioned all images using LLAVA 1.5 (Liu et al., 2024), specifically employing the LLaVA-Lightning-MPT-7B-preview model for this task. We then split the data into training and validation sets with a ratio of 100:1. Our third dataset is COCO (Lin et al., 2015), where we used the 2014 validation set to test scaling laws on an out-of-domain dataset.

### C.2   MODEL DESIGN

In this paper, we evaluate two distinct model architectures. For the PixArt model, we follow the original design presented in Chen et al. (2023). The In-Context Transformers are based on the In-Context block described in Peebles & Xie (2023). To facilitate large-scale model training, we employ QK-Norm (Dehghani et al., 2023) and RMSNorm (Zhang & Sennrich, 2019). The patch size is set to 2. Although previous work (Kaplan et al., 2020) suggests that the aspect ratio (width/depth) of Transformers does not significantly impact scaling laws, it is crucial to maintain a consistent ratio when fitting models to scaling laws. To demonstrate this, we train a series of models under a fixed computational budget, selecting models of various sizes and aspect ratios (32 and 64). We then plot the relationship between the number of parameters and loss. As illustrated in Fig. 6, mixing models with aspect ratios of 64 and 32 obscures the overall trend. To address this issue, we maintain the aspect ratio at 64 throughout.

## D   DERIVATION OF THE LIKELIHOOD

In this section, we provide a derivation of the likelihood estimation in our paper. In this paper, we use two ways to compute the likelihood. The first method is estimating the VLB (variational lower bound). Following Kingma et al. (2021); Vahdat et al. (2021), we derive a VLB in latent space. However, we cannot compute the entropy terms in the VAE. So our surrogate metric differs from the true VLB up to a constant factor.

**VAE**   The latent diffusion model (Vahdat et al., 2021; Rombach et al., 2022) consists of two components: a variational autoencoder (VAE) that encodes images into a latent space and decodes latents back into images, and a continuous diffusion model that operates in the latent space. To train the latent diffusion model, we optimize the variational encoder $q_\phi$, the decoder $p_\psi$, and the diffusion model $p_\theta$. Following Ho et al. (2020), the models are trained by minimizing the variational upper bound on the negative log-likelihood $\log P(x)$:

$$\mathcal{L}_{\theta,\phi,\psi}(x) = \mathbb{E}_{q_\phi(z_0|x)}[-\log p_\psi(x|z_0)] + KL(q_\phi(z_0|x)||p_\theta(z_0))$$
$$= \mathbb{E}_{q_\phi(z_0|x)}[-\log p_\psi(x|z_0)] + \mathbb{E}_{q_\phi(z_0|x)}[\log q_\phi(z_0|x)] + \mathbb{E}_{q_\phi(z_0|x)}[-\log p_\theta(z_0)].$$

In our implementation, we directly adopt the VAE from Stable Diffusion 1.5 and keep it fixed during training. As a result, the reconstruction term (first term) and the negative encoder entropy term (second term) remain constant across different models. In fact, the VAE in Stable Diffusion is trained following the VQGAN approach, which uses both $L1$ loss and an additional discriminator for training. Therefore, we cannot effectively estimate the reconstruction term since the decoder distribution is not tractable. To simplify further, we omit the VAE encoding process altogether. Specifically, we skip both encoding and decoding through the VAE and treat the latents produced by the VAE as the dataset samples. Under this assumption, we estimate the offset VLB directly in the latent space.

In the latent space, we model the distribution of latent variables that can be decoded into images using the VAE decoder. We denote the samples in latent space as $x$, and the noisy latent at timestep $t$ as $z_t$. The variational lower bound (VLB) in the latent space is given by Kingma et al. (2021):

$$-\log p(x) \le -\text{VLB}(x) = D_{KL}(q(z_1|x)||p(z_1)) + \mathbb{E}_{q(z_0|x)}[-\log p(x|z_0)] + \mathcal{L}_T(x),$$

where the first two terms depend only on the noise schedule, and we treat these terms as irreducible losses since the noise schedule is fixed across all models. The third term is the KL divergence between each pair of the reverse process $p(z_t|z_{t+1})$ and the forward process $q(z_t|x, z_{t+1})$:

$$\mathcal{L}_T(x) = \sum_{i=1}^{T} \mathbb{E}_{q(z_{t(i)}|x)}[D_{KL}(q||p)].$$

Since we assume that the forward and reverse processes share the same variance and both $p$ and $q$ are Gaussian distributions, the KL terms reduce to weighted $L2$ distances:

$$\mathcal{L}_T(x) = \frac{1}{2}\mathbb{E}_{\epsilon\sim\mathcal{N}(0,I)}\left[\sum_{i=1}^{T}(SNR(s) - SNR(t))\|x - x_\theta(z_t;t)\|_2^2\right],$$

where $s = t - 1$. In the limit as $T \to \infty$, the loss becomes:

$$\mathcal{L}_T(x) = -\frac{1}{2}\mathbb{E}_{\epsilon\sim\mathcal{N}(0,I)}\int_0^1 SNR'(t)\|x - x_\theta(z_t;t)\|_2^2\, dt.$$

In our case, we utilize the velocity $\mathbf{v}$ to predict the clean sample $x$ and compute the VLB.

**Normalizing Flows** Another method to compute the likelihood in our diffusion model is by viewing the diffusion process as a type of normalizing flow. Specifically, we leverage the theoretical results from Neural ODEs, which allow us to connect continuous normalizing flows with the evolution of probability density over time. In Neural ODEs, the transformation of data through the flow can be described by the following differential equation for the state variable $x_t$ as a function of time:

$$\frac{dx_t}{dt} = f_\theta(x_t, t),$$

where $f_\theta(x_t, t)$ represents the network that predicts the time-dependent vector field $\mathbf{v}$. To compute the change in log-probability of the transformed data, the log-likelihood of the input data under the flow is given by:

$$\frac{d \log p(x_t)}{dt} = -\mathrm{Tr}\left(\frac{\partial f(x_t, t)}{\partial x_t}\right),$$

where Tr represents the trace of the Jacobian matrix of $f_\theta(x_t, t)$. This equation describes how the log-density evolves as the data is pushed forward through the flow. To compute the likelihood, we integrate the following expression over the trajectory from the initial state $t_0$ to the terminal state $t_1$:

$$\log p(x_{t_1}) = \log p(x_{t_0}) - \int_{t_0}^{t_1} \mathrm{Tr}\left(\frac{\partial f_\theta(x_t, t)}{\partial x_t}\right) dt.$$

Here, $\log p(x_{t_0})$ represents the log-likelihood of the initial state (often modeled as a Gaussian), and the integral accounts for the change in probability density over time as the data evolves through the ODE. In our formulation, the network predicts the velocity $\mathbf{v}_\theta(x_t, t) = x_t' = \alpha_t' x_0 + \beta_t' \epsilon$, which corresponds to the derivative of $x_t$ with respect to time. Thus, we start with clean samples, estimate the velocity, perform an iterative reverse-time sampling, and convert the samples into Gaussian noise. We can then compute the prior likelihood of the noise easily and add it to the probability shift accumulated during reverse sampling. In our experiments, we set the steps of reverse sampling to 500 to obtain a rather accurate estimation.

# E  SCALING FLOPs COUNTING

In this section, we provide a detailed explanation of our FLOPs scaling calculations. Several prior works have employed different methods for counting FLOPs. In Kaplan et al. (2020), the authors exclude embedding matrices, bias terms, and sub-leading terms. Moreover, under their framework, the model dimension $d_{model}$ is much larger than the context length $l_{ctx}$, allowing them to disregard context-dependent terms. Consequently, the FLOPs count $N$ for their model is given by:

$$M = 12 \times d_{model} \times n_{layer} \times (2d_{attn} + d_{ff}), \tag{9}$$

where $d_{model}$ represents the model dimension, $n_{layer}$ denotes the depth of the model, $d_{attn}$ refers to the attention dimension, and $d_{ff}$ represents the feed-forward layer dimension.

In contrast, Hoffmann et al. (2022) includes all training FLOPs, accounting for embedding matrices, attention mechanisms, dense blocks, logits, and context-dependent terms. Specifically, their FLOP computation includes:

- **Embedding**: $2 \times l_{ctx} \times N_{voc} \times d_{model}$
- **Attention**:
  - **QKV Mapping**: $2 \times 3 \times l_{ctx} \times d_{model} \times d_{model}$
  - **QK**: $2 \times l_{ctx} \times l_{ctx} \times d_{model}$
  - **Softmax**: $3 \times n_{head} \times l_{ctx} \times l_{ctx}$
  - **Mask**: $2 \times l_{ctx} \times l_{ctx} \times d_{model}$
  - **Projection**: $2 \times l_{ctx} \times d_{model} \times d_{model}$
- **Dense**: $2 \times l_{ctx} \times (d_{model} \times d_{ff} \times 2)$
- **Logits**: $2 \times l_{ctx} \times d_{model} \times N_{voc}$

Further details can be found in the Appendix **F** of Hoffmann et al. (2022).

In Bi et al. (2024), the authors omit the embedding computation but retain the context-dependent terms, which aligns with our approach. The parameter scaling calculation for vanilla Transformers in this paper follows the same format as theirs. We now present detailed scaling FLOPs calculations for the In-Context Transformers and Cross-Attn Transformers used in our experiments.

Attention blocks are the primary components responsible for scaling in Transformer architectures. In line with previous studies, we only consider the attention blocks, excluding embedding matrices and sub-leading terms. Unlike large language models (LLMs), our model dimension is comparable to the context length, and therefore, we include context-dependent terms. In this section, we present

FLOPs per sample rather than parameters, as different tokens participate in different parts of the cross-attention computation. Additionally, since our input length is fixed, the FLOPs per sample are straightforward to compute.

**In-Context Transformers** In-Context Transformers process a joint embedding consisting of text, image, and time tokens, all of which undergo attention computation. Tab. 3 details the FLOPs calculations for a single attention layer.

Table 3: Scaling FLOPs Calculation in In-Context Transformers

| Operation | FLOPs per Sample |
|---|---|
| Self-Attn: QKV Projection | $3 \times 2 \times n_{layer} \times l_{ctx} \times d_{model} \times 3 \times d_{attn}$ |
| Self-Attn: QK | $3 \times 2 \times n_{layer} \times l_{ctx} \times l_{ctx} \times d_{attn}$ |
| Self-Attn: Mask | $3 \times 2 \times n_{layer} \times l_{ctx} \times l_{ctx} \times d_{attn}$ |
| Self-Attn: Projection | $3 \times 2 \times n_{layer} \times l_{ctx} \times d_{model} \times d_{attn}$ |
| Self-Attn: FFN | $3 \times 2 \times 2 \times n_{layer} \times l_{ctx} \times 4 \times d_{model}^2$ |

In our experiments, we set $d_{model} = d_{attn}$, and $l_{ctx} = 377$, where $l_{ctx} = l_{img}(256) + l_{text}(120) + l_{time}(1)$. Thus, the simplified FLOPs-per-sample scaling equation $M$ is:

$$M = 72 \times l_{ctx} \times n_{layer} \times d_{model}^2 + 12 \times n_{layer} \times l_{ctx}^2 \times d_{model} \tag{10}$$

**Cross-Attn Transformers** In Cross-Attn Transformers, each attention block consists of self-attention and cross-attention mechanisms to integrate text information. The cross-attention uses image embeddings as the query and text embeddings as the key and value. The attention mask reflects the cross-modal similarity between image patches and text segments. As a result, the FLOPs calculation differs from that of models using joint text-image embeddings. Tab. 4 lists the FLOPs costs for each operation.

Table 4: Scaling FLOPs Calculation in Cross-Attn Transformers

| Operation | FLOPs per Sample |
|---|---|
| Self-Attn: QKV Projection | $3 \times 2 \times n_{layer} \times l_{img} \times d_{model} \times 3 \times d_{attn}$ |
| Self-Attn: QK | $3 \times 2 \times n_{layer} \times l_{img} \times l_{img} \times d_{attn}$ |
| Self-Attn: Mask | $3 \times 2 \times n_{layer} \times l_{img} \times l_{img} \times d_{attn}$ |
| Self-Attn: Projection | $3 \times 2 \times n_{layer} \times l_{img} \times d_{model} \times d_{attn}$ |
| Cross-Attn QKV | $3 \times 2 \times n_{layer} \times (l_{img} + 2 \times l_{text}) \times d_{model} \times d_{attn}$ |
| Cross-Attn QK | $3 \times 2 \times n_{layer} \times l_{text} \times l_{img} \times d_{attn}$ |
| Cross-Attn Mask | $3 \times 2 \times n_{layer} \times l_{text} \times l_{img} \times d_{attn}$ |
| Cross-Attn Projection | $3 \times 2 \times n_{layer} \times l_{img} \times d_{model} \times d_{attn}$ |
| FFN | $3 \times 2 \times 2 \times n_{layer} \times l_{img} \times 4 \times d_{model}^2$ |

Based on the experimental settings, we can simplify the FLOPs calculation as follows:

$$M = 84 \times n_{layer} \times l_{img} \times d_{model}^2 + 12 \times n_{layer} \times l_{img}^2 \times d_{attn}$$
$$+ 12 \times n_{layer} \times l_{text} \times d_{model}^2 + 12 \times n_{layer} \times l_{text} \times l_{img} \times d_{model} \tag{11}$$

**Context-Dependent Terms** From the equations above, it is evident that some context-dependent terms, such as $12 \times n_{layer} \times l_{ctx}^2 \times d_{model}$, cannot be omitted. In our experiments, the aspect ratio of

Transformers (width/depth=64) is maintained across all model sizes. The context length $l_{ctx}$ is 377 (image: 256, text: 120, time: 1), and $d_{model} = n_{layer} \times 64$. Since $l_{ctx}$ and $d_{model}$ are comparable, the context-dependent terms must be retained.

# F  ABLATIONS

## F.1  DIFFUSION FORMULATION

In diffusion models, various formulations for noise schedules, timestep schedules, and prediction objectives have been proposed. These three components are interdependent and require specific tuning to achieve optimal performance. In this paper, we explore several common formulations and conduct ablation studies to identify the best combination in our setting.

Below, we list the candidate formulations used in our ablation study.

**Noise Schedule**

**Discrete Diffusion Models**

**DDPM**  Denoising Diffusion Probabilistic Models (DDPM) (Ho et al., 2020) is a discrete-time diffusion model that generates noisy samples via the following formula:

$$x_t = \alpha_t x_0 + \beta_t \epsilon$$

where $\epsilon$ is Gaussian noise, and $\alpha_t$ and $\beta_t$ satisfy $\alpha_t^2 + \beta_t^2 = 1$. Given a sequence of $\sigma_t$, the scaling factor can be defined as:

$$\alpha_t = \sqrt{\prod_{s=0}^{t} (1 - \sigma_t)} \tag{12}$$

In DDPM, $\sigma_t$ follows:

$$\sigma_t = \sigma_0 + \frac{t}{T}(\sigma_T - \sigma_0) \tag{13}$$

**LDM**  Latent Diffusion Models (LDM) (Rombach et al., 2022), as used in Stable Diffusion, is a variant of the DDPM schedule. It is also a variance-preserving formulation, sharing the same structure as DDPM but employing a different noise schedule:

$$\sigma_t = \left( \sqrt{\sigma_0} + \frac{t}{T} \left( \sqrt{\sigma_T} - \sqrt{\sigma_0} \right) \right)^2$$

**Continuous Diffusion Models**

**VP**  Variance Preserving (VP) diffusion (Song et al., 2021b) is the continuous counterpart of DDPM, where the noise is added while preserving variance across timesteps. The sampling process is given by:

$$x_t = e^{-\frac{1}{2} \int_0^t \sigma_s ds} x_0 + \sqrt{1 - e^{-\int_0^t \sigma_s ds}} \epsilon$$

where $t \in [0, 1]$.

**Rectified Flow**  Rectified Flow (RF) (Liu et al., 2022; Lipman et al., 2022; Albergo et al., 2023) is another continuous-time formulation, where a straight-line interpolation is defined between the initial sample $x_0$ and the Gaussian noise $\epsilon$. The process is described by:

$$x_t = (1 - t)x_0 + t\epsilon$$

**Prediction Type**

**Noise Prediction ($\epsilon$)**   The network predicts the Gaussian noise $\epsilon \sim \mathcal{N}(0, I)$ added to the samples during the diffusion process.

**Velocity Prediction (v)**   In this formulation, the network predicts the velocity $\mathbf{v}(x_t, t)$, which is defined as the derivative of the noisy sample $x_t$ with respect to time. If the noisy sample $x_t$ is defined by:

$$x_t = \alpha_t x_0 + \beta_t \epsilon$$

the velocity is given by:

$$\mathbf{v}(x_t, t) = x'_t = \alpha'_t x_0 + \beta'_t \epsilon$$

where $\alpha'_t$ and $\beta'_t$ are the derivatives of $\alpha_t$ and $\beta_t$ with respect to timestep $t$.

**Score Prediction (s)**   The network predicts the score function $\mathbf{s}(x, t) = \nabla \log P(x, t)$, which is the gradient of the log-probability density function. The score can be derived as:

$$\mathbf{s}(x, t) = -\frac{\epsilon}{\beta_t}$$

**Timestep Sampling Schedule**

**Uniform Timestep Schedule**   In this schedule, the timestep $t$ is uniformly sampled. For discrete-time diffusion models:

$$t \sim \mathcal{U}(0, 1, 2, \ldots, 999)$$

For continuous-time diffusion models:

$$t \sim \mathcal{U}(0, 1)$$

**Logit-Normal (LN) Timestep Schedule**   The Logit-Normal (LN) timestep schedule $\pi_{ln}(t; m, s)$, proposed in Esser et al. (2024), generates timesteps according to the following distribution:

$$\pi_{ln}(t; m, s) = \frac{1}{s\sqrt{2\pi}} \cdot \frac{1}{t(1-t)} \exp\left(-\frac{(logit(t) - m)^2}{2s^2}\right),$$

where $logit(t) = \log\left(\frac{t}{1-t}\right)$. The LN schedule has two parameters: $m$ and $s$. It defines an unimodal distribution, where $m$ controls the center of the distribution in logit space, shifting the emphasis of training samples towards noisier or cleaner regions. The parameter $s$ adjusts the spread of the distribution, determining its width. As suggested in Esser et al. (2024), to obtain a timestep, we first sample $u \sim \mathcal{N}(m, s)$, and then transform it using the logistic function. For discrete-time diffusion, after obtaining $t \sim \pi_{ln}(t; m, s)$, we scale $t$ by $t = \text{round}(t \times 999)$ to obtain a discrete timestep. Following Esser et al. (2024), we utilized the parameters $m = 0.0, s = 1.00$ and didn't sweep over $m$ and $s$. More details and visualizations can be found in Esser et al. (2024) Appendix **B.4**.

We conducted a series of experiments using different combinations of formulations. Selective combinations are listed in Tab. 5. A '$-$' indicates that the combination is either not comparable with other formulations or that training diverges. We assume that the choice of formulation will not be significantly affected by specific model designs or datasets. All experiments were conducted using Pixart (Chen et al., 2023), a popular text-to-image diffusion transformer architecture. Specifically, we used a small model with 12 layers and a hidden size of 384, setting the patch size to 2. The models were trained on JourneyDB (Sun et al., 2023), a medium-sized text-to-image dataset consisting of synthetic images collected from Midjourney. All models were trained for 400k steps using AdamW as the optimizer. As shown in Tab. 5, the optimal combination is **[RF, LN, v]**, achieving an FID of **36.336** and a Clip Score of **0.26684**. This combination achieved the best performance on both metrics and therefore, we used this setting in the remaining experiments.

F.2   EMA

The Exponential Moving Average (EMA) coefficient is crucial to the loss curve and determining the final results. In EMA, the loss $l$ is updated as $l = (1 - \alpha_{\text{EMA}})l + \alpha_{\text{EMA}}v$, where $v$ represents the latest loss value. EMA smooths the cumulative loss and reduces fluctuations. However, applying EMA can lead to an overestimation of the loss during the early stages of training. As illustrated in Fig. 7,

| Noise Schedule | Timestep Schedule | Prediction Type | FID | CLIP Score |
|---|---|---|---|---|
| DDPM | Uniform | $\epsilon$ | 40.469 | 0.26123 |
| DDPM | Uniform | v | 74.049 | 0.23136 |
| DDPM | LN | $\epsilon$ | 40.100 | 0.26283 |
| LDM | Uniform | $\epsilon$ | 39.001 | 0.26423 |
| LDM | Uniform | v | — | — |
| LDM | LN | $\epsilon$ | 38.196 | 0.26624 |
| VP | Uniform | $\epsilon$ | 44.343 | 0.26148 |
| VP | Uniform | v | 41.808 | 0.26320 |
| VP | Uniform | s | 45.808 | 0.25970 |
| VP | LN | $\epsilon$ | 42.872 | 0.26354 |
| VP | LN | v | 44.107 | 0.26380 |
| RF | Uniform | v | 44.840 | 0.25682 |
| RF | Uniform | s | — | — |
| RF | Uniform | $\epsilon$ | — | — |
| RF | LN | v | **36.336** | **0.26684** |

Table 5: Ablation on diffusion formulations.

a larger EMA coefficient results in a higher loss compared to the actual value, which may introduce significant bias in scaling curve fitting and, consequently, lead to inefficient use of computational resources. From Fig. 7, we observe that $\alpha_{\text{EMA}} = 0.9$ is the optimal choice, as it effectively smooths the loss curve while only slightly inflating the values during the initial phase of training.

### F.3 CLASSIFIER-FREE GUIDANCE & SAMPLING STEPS

We perform ablation studies on CFG scales and the number of steps using the compute-optimal models trained with a budget of 6e18. In Fig. 8, we evaluate several CFG scales (2.5, 5.0, 7.5, 10.0) and compute FID across different step counts. We found that 25 steps are sufficient to achieve good results. Next, we fix the number of steps at 25 and evaluate performance across different CFG scales. As shown in Fig. 9, a CFG of 10.0 yields the best results, and we select this configuration as the default.

## G REBUTTAL

In this section, we add extra experiment results and discussions in response to the reviewers.

### G.1 FLUX

Reviewer 2p8z and CvMz raised their concern about whether scaling laws can be observed in modern DiT architectures. To address the reviewers' concerns, we test on FLUX. As shown in Fig. 10, the loss curve clearly shows a linear relation, which shows that the scaling laws exist.

### G.2 PIXART

To further address Reviewer 2p8z and CvMz's concern about whether scaling laws can be observed in modern DiT architectures. We conduct experiments on Pixart. As shown in Fig. 11, the loss curve also shows a linear relation, which further supports that the existence of scaling laws doesn't rely on model architectures.

### G.3 RESOLUTION 512

Reviewer CVDQ, CvMz, and 2p8z are concerned with whether the existence of scaling laws depends on a specific dataset and whether different resolution matters. We perform several experiments on a

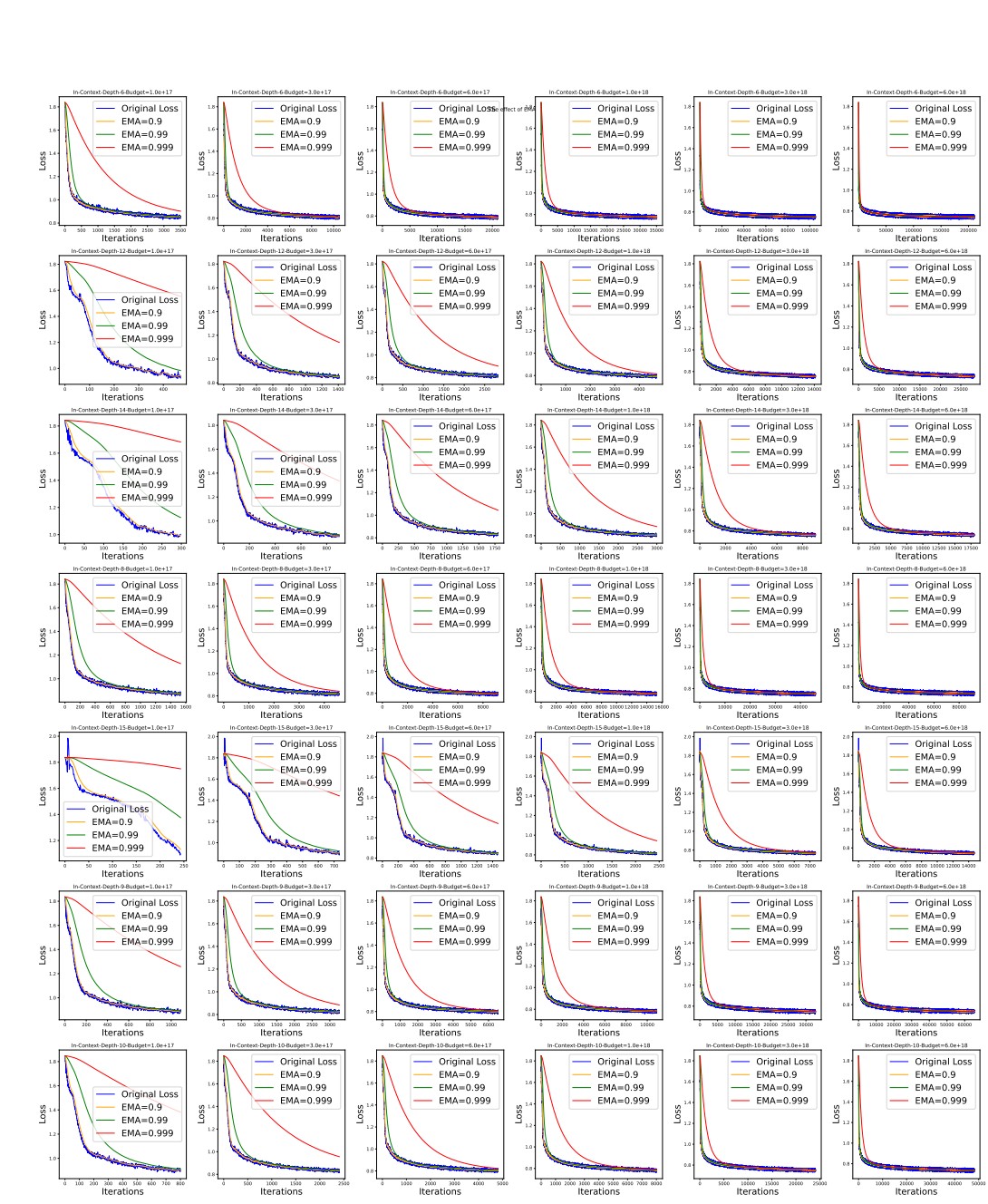

Figure 7: The effect of EMA.

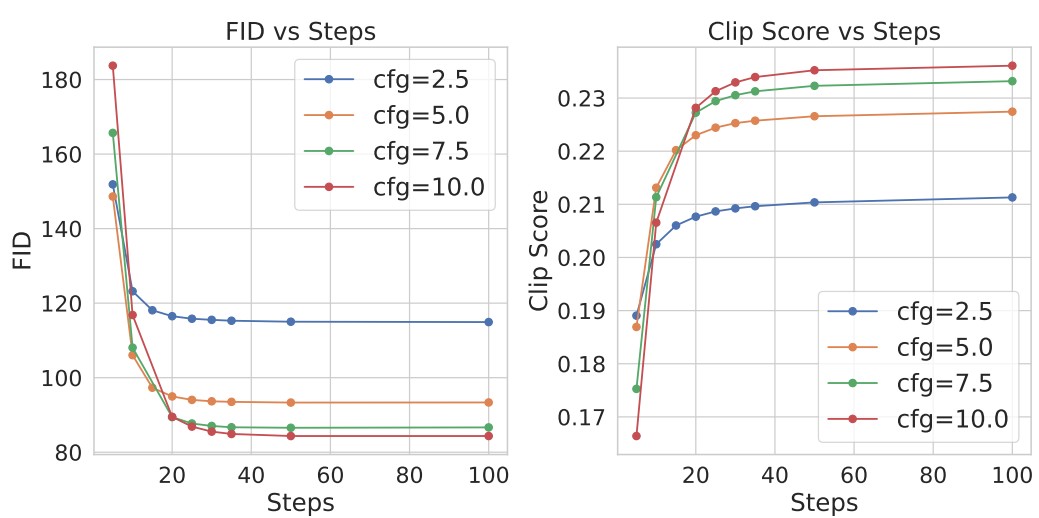

Figure 8: Ablation on sampling steps

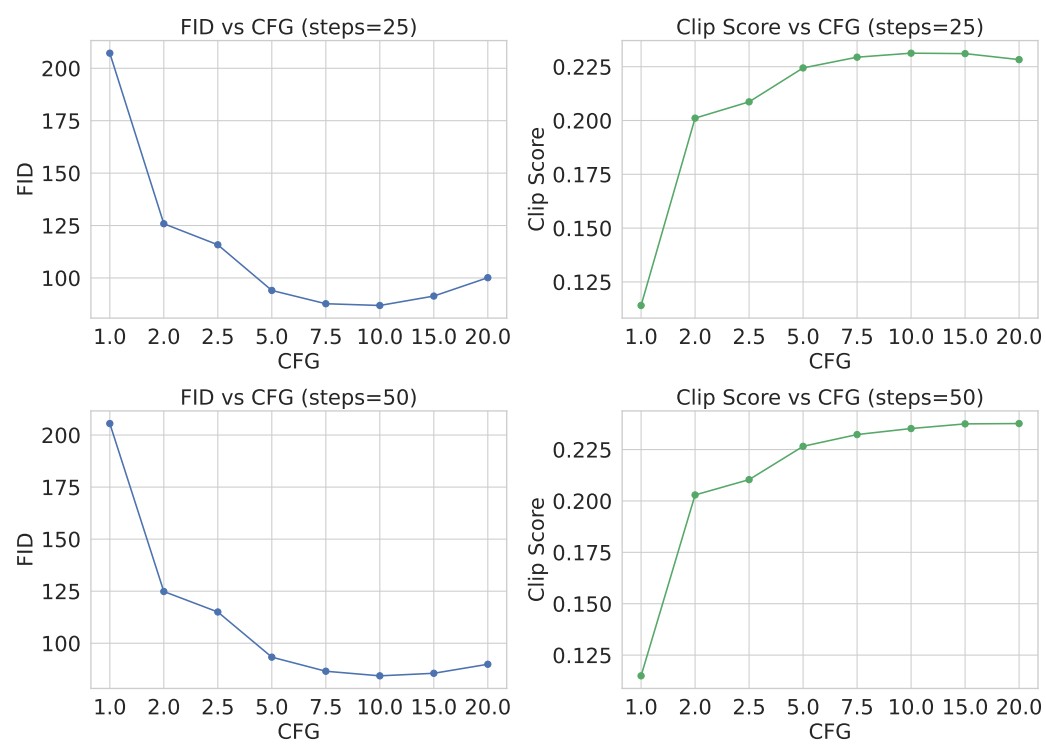

Figure 9: Ablation on cfg scale

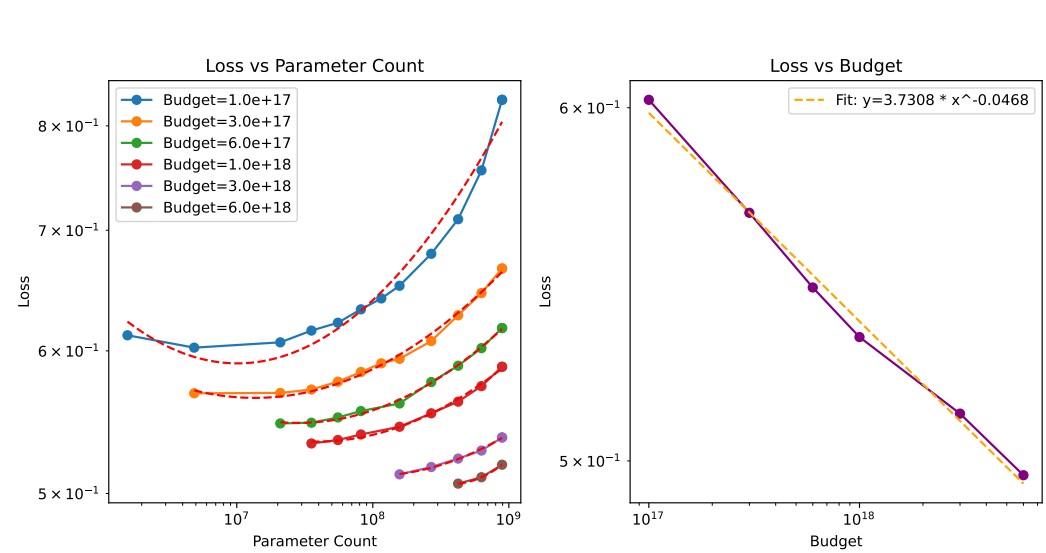

Figure 10: Scaling laws for FLUX

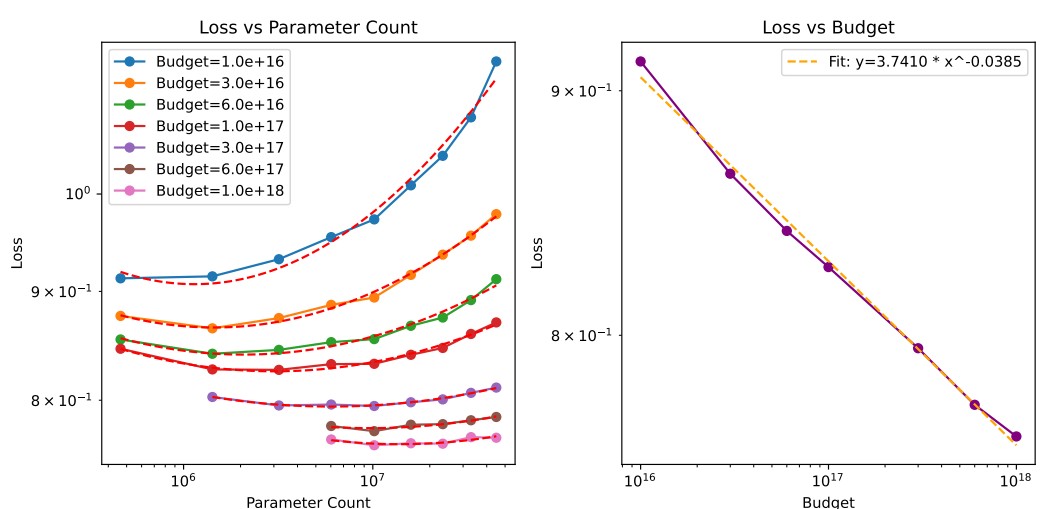

Figure 11: Scaling laws for Pixart

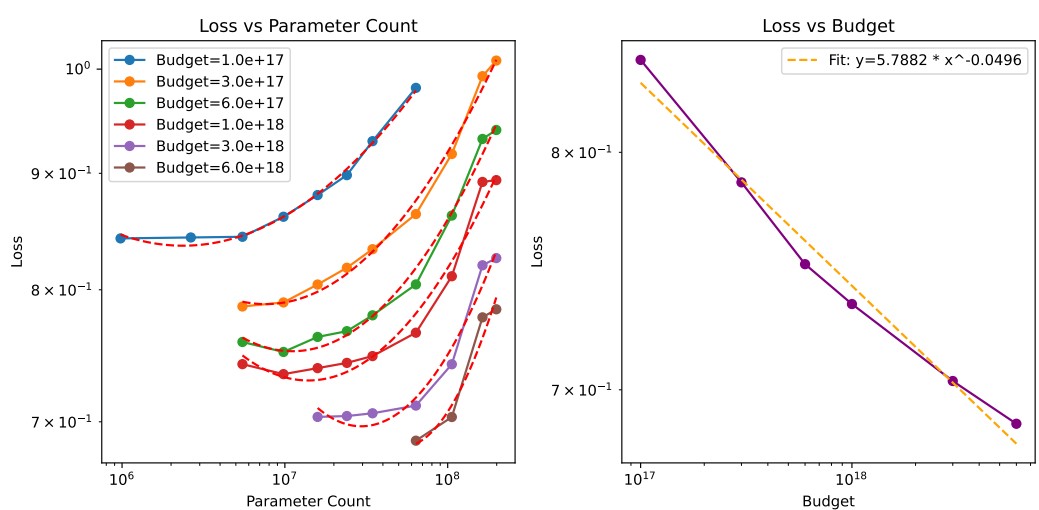

Figure 12: Scaling laws for 512 Resolution.

dataset with images at 512 resolutions. The curve under each budget can be well fitted by a parabola and the relation between loss and budgets shows clear linear trends. The results in Fig. 12 indicate that resolution and data will not affect the existence.

### G.4    MASKED DIFFUSION TRANSFORMERS

We add the discussion for missing references pointed out by reviewer ssc5.

Masked Diffusion Transformers (v1, v2) Gao et al. (2023) are the first works to explore mask latent modeling into diffusion training. It introduces masks into the training to enhance the ability to understand contextual relations.

### G.5    LEARNING RATE SCAN

Reviewer 2p8z expressed concerns about the impact of learning rate. To address this, we tested the sensitivity of learning rate using the largest models from our experiments. The results, presented in Fig. 13, show that increasing the learning rate does not lead to improved loss. Our findings indicate that 1e-4 is generally a robust and effective choice.

### G.6    DATA ASSESSMENT

To demonstrate how scaling laws can be used to assess data quality, we applied them to an additional dataset. This dataset contains the same images as the one used in our main paper but features sparse tag captions instead of dense descriptions. The scaling curve for FID is shown in Fig. 14, with an exponent of -0.216. In contrast, the scaling exponent in the main text is -0.234, indicating that the FID for the dataset in the main text decreases more rapidly, leading to better performance. This comparison suggests that long and dense captions provide a significant advantage over sparse tag captions in terms of data quality.

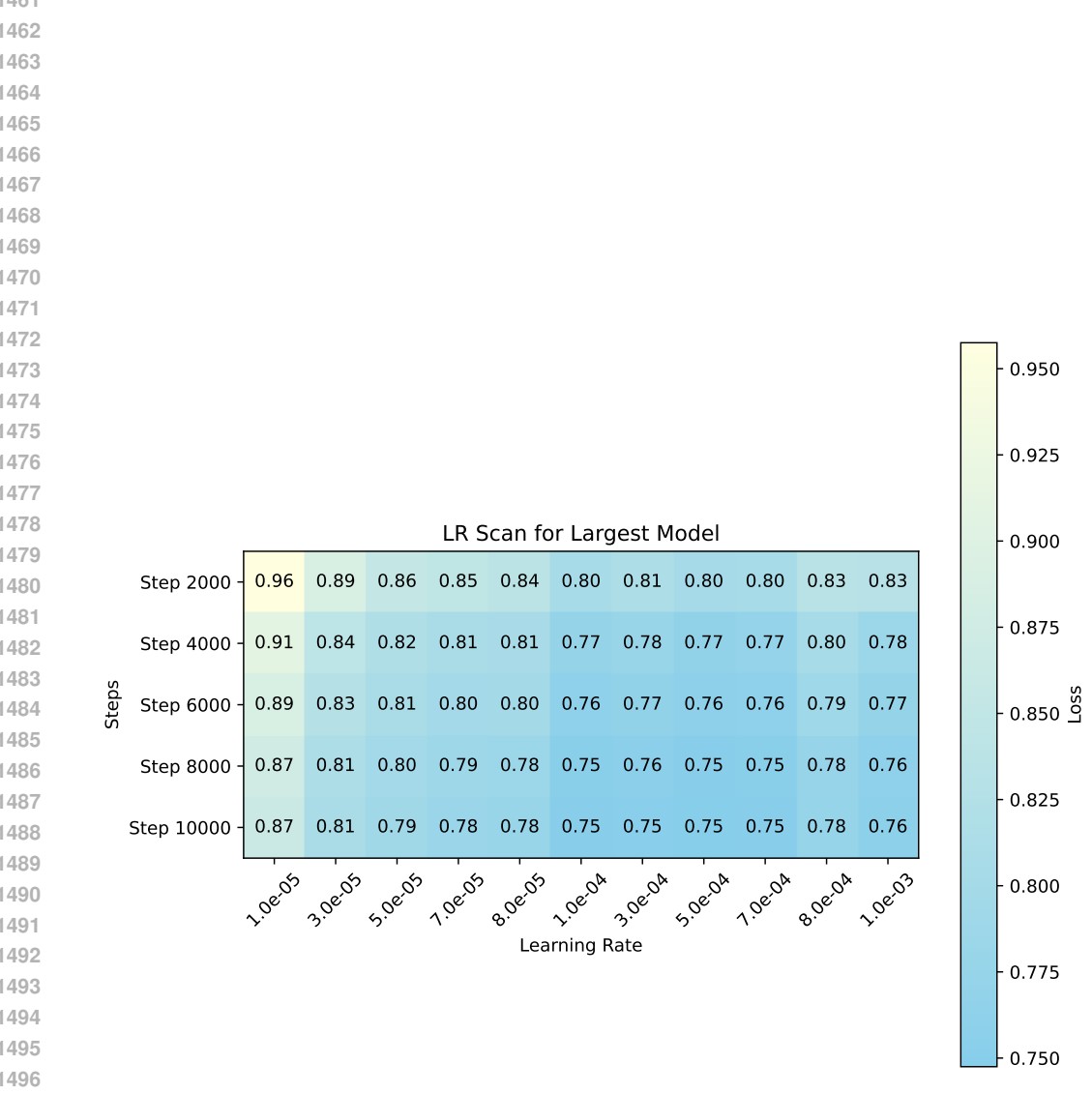

Figure 13: Learning rate scan for the largest models.

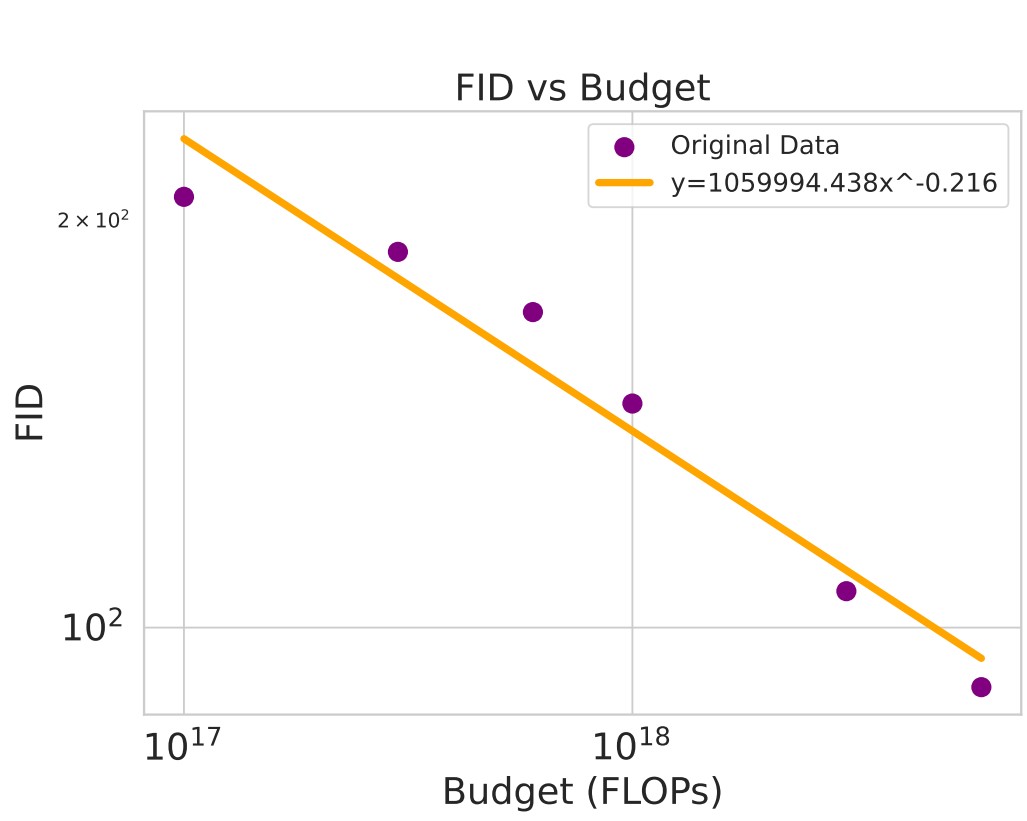

Figure 14: Scaling curves for FID on Sparse-captioned dataset.