# OpenReview forum: "Scaling Laws for Diffusion Transformers"
_ICLR.cc/2025/Conference — Submitted to ICLR 2025_

### Official Review · Reviewer_CvMz · 2024-10-27

**Soundness:** 2
**Presentation:** 3
**Contribution:** 3
**Rating:** 6
**Confidence:** 4

**Summary:**

This work tries to reveal the scaling law of Diffusion Transformer in the field of text-to-image generation. It is based on in-context dit and rectified flow matching, and performs a series of experiments between 1e17 and 6e18 flops. By this, it finds the power law relationship between the objective function, model parameters and compute. It also uses the loss of a 1B dit at the compute of 1e21 flops to confirm the correctness of the equation. At last, it conducts a comparative test between in-context and cross-attention dit, to show that the scaling law can help to indicate the effectiveness of the model architecture's design.

**Strengths:**

1. This work first proposes the scaling law of Diffusion Transformer in text-to-image generation. And it gets several formulated equations between loss, parameters and compute, which is confirmed by an additional experiment.
2.  It shows that different metrics (loss, VLB, likelihood) has similar trends with regard to compute budget. And the performance of generation measured by FID does, too.
3.  It indicates the application of the scaling law: serving as a benchmark for measuring the effectiveness of model architectures.

**Weaknesses:**

1. The range of compute budget for experiments is a bit narrow. The count of data points is a little few for drawing a curve. According to "Scaling Laws for Autoregressive Generative Modeling"[1], they did experiments from 1e14 to 1e23 flops.
2. This work's experiment is not as sufficient as the scaling law for AR[1]. Maybe it needs some curves like [1].
3. The base setting of model architecture is not good. Recently, popular settings of in-context dit mainly refer to SD3[2] and FLUX, which only concat text and image as input but use timesteps for adaptive layernorm. This setting is important and may lead to suboptimal conclusion.
4. It said that "scaling laws can serve as a reliable and predictable benchmark for evaluating the effectiveness of both model architectures and datasets". But it misses experiments to show how to use scaling law to choose training datasets.
5. Did not indicate the resolution of training datasets. Different resolutions may affect the conclusion.

[1] Scaling Laws for Autoregressive Generative Modeling
[2] Scaling Rectified Flow Transformers for High-Resolution Image Synthesis

**Questions:**

1. Some images in the paper only have one or two curves. Does this imply that it is already experimenting with the optimal model parameters calculated by the scaling law?
2. Why don't you use bigger models for base experiments? Common diffusion models, such as SD3 and FLUX, has 10B parameters or so. Only doing experiments with models less than 1B may is not general enough.

---

> ### Author Response · Authors · 2024-11-27
> **Response to reviewer CvMz**
>
> We thank the reviewer for detailed comments and feedback. We respond to your concern related to our paper in the following part.
>
> **Q1: Narrow compute budget range.**
>
> A1: Compared to [1], the range of our compute budget is indeed smaller. However, as referenced in [3], spanning nearly three orders of magnitude is sufficient to fit accurate scaling laws. Additionally, as demonstrated in our experiments, larger computing budgets require significantly more computational resources. For example, training a model corresponding to a compute budget of 1e23 would likely require over 10 billion parameters, which is beyond the scope of a typical research project due to the excessive computational demands. A broader range of compute range will result in more points for scaling laws estimation and yield more accurate estimation. However, it will not affect the existence of the scaling laws and the correctness of our methods on how to estimate scaling laws and how to use them.
>
> **Q2: Insufficient experiments compared with [1]**
>
> A2: We acknowledge that the experiments in our paper are not as extensive as those in [1]. The primary focus of our work is to demonstrate the existence of scaling laws for diffusion transformers and provide a guideline on how to predict the scaling laws and utilize the scaling laws. The experiments in our paper are sufficient to support these claims.
> Additionally, we have supplemented our study with experiments on different model architectures (Flux **Supp Section G.1**, Pixart **Supp Section G.2**), image resolutions (**Supp Section G.3**), and captions (**Supp Section G.4**), showing that the existence of scaling laws does not depend on specific transformer designs or data configurations. While [1] includes numerous experiments that provide a more comprehensive view of scaling laws, these additional studies do not affect the fundamental conclusion of the scaling laws’ existence and the guideline, which is the core focus of our paper. We plan to explore these aspects further in future work.
>
> **Q3: Some images in the paper only have one or two curves. Does this imply that it is already experimenting with the optimal model parameters calculated by the scaling law?**
>
> A3: Yes, some figures indeed use the optimal models at each budget to create the curves. This is because the purpose of each figure varies, and the visualized model results differ accordingly. In Figure 1 (left), we test models of different sizes under each compute budget to determine the optimal model size and token number for that budget. So, we plot the results of all models and pick the optimal one. For Figures 2 and 3, our goal is to predict the optimal performance at each budget. The optimal performances are achieved by the optimal models. Therefore, the results shown correspond to the loss achieved by the optimal model at each budget, and only these values are plotted.
>
> **Q4: Model architecture.**
>
> A4: First, based on prior research on large language models (LLMs)[1](Figure 5.), the existence of scaling laws for transformer-based models is generally independent of the specific architectural design. While the model architecture may influence the coefficients and exponents of the scaling laws, it does not affect their existence.
> To address the reviewer’s concern, we conducted a series of experiments to validate our conclusions. In the main text, we have already demonstrated this through experiments on PixArt [2], and we further provide predictions of the loss in **Supplementary Section G.2**. Additionally, we extended our validation to the latest DiT architecture by evaluating its performance under the Flux budget. The results are included in **Supplementary Section G.1** as well. In both cases, as shown in the figures provided, there is a clear linear relationship between the compute budget and the loss, which strongly supports the presence of scaling laws.
>
> **Q5: Resolution**
>
> A5: All experiments in our paper use images with a resolution of [256, 256].
> To address the reviewer’s concern regarding resolution, we also conducted experiments using images with a resolution of 512. The results are presented in **Supplementary Section G.3**. From the figure, we observe that at each training budget, the performance of models with varying sizes can be accurately fitted using a parabola. Moreover, the loss of the optimal model under each training budget also follows a linear relationship, further confirming the existence of scaling laws.

---

> ### Author Response · Authors · 2024-11-27
> **Response to reviewer CvMz**
>
> **Q6: Model size**
>
> A6:
> Regarding the model size, our experiments focus on models around 1B parameters. This aligns with the majority of open-sourced diffusion models, which are typically under 10B parameters, placing our model within the same order of magnitude. The primary goal of our paper is not to train a SOTA-sized diffusion model but rather to demonstrate the existence of scaling laws for diffusion models and provide a guideline on how to compute and utilize the scaling laws.
>
> To address the reviewer’s concerns, we suggest drawing a parallel to the development of scaling laws in the LLM domain. Scaling laws are a developing field, and the exploration of their existence and limitations has historically been incremental. For example, the initial verification of scaling laws for LLMs was conducted with models at the 600M parameter scale [1]. Over time, this expanded to tens of billions of parameters and eventually to the current 400B or more scale [5, 6]. Notably, when the first paper on scaling laws for LLM[1] was published, the largest model, GPT-3[7], had 175B parameters, creating a size ratio of **175B/0.6B ≈ 291** between SOTA models and the scaling law benchmarks.
>
> According to most reviewers’ comments and the best of our knowledge, our work is the first work to verify the existence of scaling laws in Diffusion Transformers. Most open-sourced diffusion models today are below 10B parameters, and our 1B-scale models are much closer to SOTA in this domain compared with the first few works in LLM (size ratio 291), falling within the same order of magnitude. Scaling law validation is a step-by-step process, and the scale of models used increases over time as the field progresses.
>
> Besides, training a 3B diffusion model would require approximately 676B tokens and close to a month of training, while a 10B model would require 2.2T tokens and several months of training. This level of resource demand far exceeds the affordance of a research project. Our work focuses on proving the existence of scaling laws and providing a guideline, not on training SOTA diffusion models. Larger-scale scaling law validation, like in the LLM field, will require further exploration in future studies.
>
> **Q7: Data assessment**
>
> A7: We conducted several experiments on a dataset with sparse captions to demonstrate how scaling laws can be used to assess data quality. The images in this dataset are identical to those in the dataset used in our main study, with the captions being the only variable. Following the procedure outlined in the main paper, we fit scaling laws for FID on the sparse-captioned dataset. The results are presented in **Supplementary Material Section G.6** of the supplementary materials, where the scaling laws exhibit a clear linear trend. The equation for the fitted line is 1.059 * 10^6 * x^{-0.216}.
> To compare data quality, we analyzed the scaling exponents. In the main text, the exponent for the dense-captioned dataset is -0.234, which is lower than -0.216. This indicates that the FID for the dense-captioned dataset decreases more rapidly, reflecting better performance. Thus, we conclude that dense captions are superior to sparse captions, consistent with our empirical observations.
>
> [1] Scaling Laws for Neural Language Models. Kaplan et al. https://arxiv.org/abs/2001.08361
>
> [2] PixArt-α: Fast Training of Diffusion Transformer for Photorealistic Text-to-Image Synthesis. Chen et al. https://arxiv.org/abs/2310.00426
>
> [3] Training Compute-Optimal Large Language Models. Hoffmann et al. https://arxiv.org/abs/2203.15556
>
> [4] DeepSeek LLM: Scaling Open-Source Language Models with Longtermism. Bi et al. https://arxiv.org/abs/2401.02954
>
> [5] GPT-4 Technical Report. OpenAI. https://arxiv.org/abs/2303.08774
>
> [6] The Llama 3 Herd of Models. Dubey et al. https://arxiv.org/abs/2407.21783
>
> [7] Language Models are Few-Shot Learner. Brown et al. https://arxiv.org/abs/2005.14165

---

> > ### Comment · Reviewer_CvMz · 2024-11-30
> >
> > Your answer solved most of my questions, but weakness3 is mainly questioning that your conclusion that cross attn is better than In-context is problematic. Because your setting of In-context is not common.

---

> ### Author Response · Authors · 2024-11-29
>
> Dear Reviewer CvMz,
>
> We have responded to your concerns and would be happy to clarify further if needed. Please let us know if you have any additional questions or points for discussion.
>
> Thank you for your time and feedback. Wishing you a great day!
>
> Best regards,
>
> The Authors

---

> ### Author Response · Authors · 2024-11-30
>
> Thank you for your valuable feedback. We appreciate the reviewer’s concerns and agree that the conclusions presented in our paper require refinement. Specifically, we acknowledge that our statement suggesting **"in-context learning is worse than cross-attention"** is problematic and stems from imprecise wording. To clarify, the in-context method used in our study should be referred to as **vanilla in-context**, which concatenates all images, text, and time tokens and processes them using standard transformers. This approach is distinct from more advanced methods such as FLUX or MMDIT, as correctly highlighted by the reviewer.
>
> We recognize that our language in the paper was not sufficiently precise. The appropriate conclusion should state that **vanilla in-context learning is less effective than cross-attention in terms of scalability**, rather than broadly asserting that in-context learning is inferior to cross-attention. This correction will be made in future revisions.
>
> Another issue is related to the wording used in the comparison presented in Section 4, which focuses on the **scalability** of different model architectures. By analyzing the coefficients of the FID and loss scaling curves, our intent was to evaluate which architecture performs better when **scaled up**. However, we acknowledge that the conclusions may appear oversimplified. For instance, as shown in Figure 5 of our paper, there is a crossing point where performance changes depending on the model architecture and available training budget. Without specifying training budgets, such conclusions can be misleading. Generally, as we increase training budgets—a common scenario of interest—cross-attention outperforms vanilla in-context learning. This finding aligns with results from both the original DiT paper [1] and PixArt [2], which demonstrate that cross-attention achieves superior performance when trained with very large budgets. We will address and clarify this in subsequent versions of our paper.
>
> We also agree with the reviewer that modern in-context learning architectures, such as FLUX or MMDIT, may perform better. However, comparing these advanced designs was outside the scope of our current study. Vanilla in-context learning and FLUX/MMDit in-context learning differ substantially in their architectural designs, including dual-branch versus single-branch structures, additional text conditions, and other advanced techniques. The design space for DiT remains vast and continues to evolve. For instance, while PixArt [2] illustrates the advantages of cross-attention over vanilla in-context learning, advanced in-context approaches like MMDIT (SD3) and FLUX have shown improved performance. Looking forward, it is possible that future iterations of DiT will see enhanced cross-attention designs reclaiming the advantage. Therefore, at this stage, we believe that comparisons should focus on **specific model architectures**, evaluated individually. A definitive conclusion about the relative merits of in-context learning versus cross-attention is premature.
>
> In Section 4 of our paper, our goal was to demonstrate how scaling laws can be used to assess **the scalability of different architectures** on a case-by-case basis. The conclusions drawn are specific to the architectures analyzed—namely, that **vanilla in-context learning is less effective than cross-attention when scaling up**—and should not be interpreted as a general critique of in-context mechanisms. We will ensure that this distinction is clearly expressed in future revisions.
>
>
> [1] Scalable Diffusion Models with Transformers. Peebles et al. https://arxiv.org/abs/2212.09748
>
> [2] PixArt-α: Fast Training of Diffusion Transformer for Photorealistic Text-to-Image Synthesis. Chen et al. https://arxiv.org/abs/2310.00426

---

> > ### Comment · Reviewer_CvMz · 2024-11-30
> >
> > I also read the author's reply to other reviewers. The author's additional experiments solved most of my concerns. Regarding the not completely accurate conclusion mentioned above, I hope the author will pay attention to it. It is likely to cause misunderstanding among readers. I decided to give it 6 points.

---

> > > ### Author Response · Authors · 2024-12-01
> > >
> > > We will pay attention to the potentially misleading conclusion. We thank the reviewer for constructive feedback and we will fix the above issues in later version.

---

### Official Review · Reviewer_CVDQ · 2024-11-02

**Soundness:** 2
**Presentation:** 3
**Contribution:** 2
**Rating:** 5
**Confidence:** 3

**Summary:**

This work studies the scaling laws of diffusion transformer for text-to-image generation. It tries to unveil the relationship between the training budget and the optimal model size/training data quantity. Under careful designed experimental settings, the disclosed scaling laws served as a predictor for the model size/performance under given a specific training budget.

**Strengths:**

This is the first study of scaling laws in DiT based text-to-image generation models. It disclosed some relationships between training budget and model size/training data quantity under a specific setting. The results might be useful for the community when designing future DiT based text-to-image generation.

**Weaknesses:**

Experiment results are somewhat limited. All experiments were done with a very small dataset and small models (<= 1B parameters). These settings are quite different from many DiT based text-to-image models in terms of data quantity and model size. Whether the conclusion will hold for different setting is unknown to us.

**Questions:**

please refer to the weakness

---

> ### Author Response · Authors · 2024-11-27
> **Response to reviewer CVDQ**
>
> We thank the reviewer for detailed comments and feedback. We respond to your concern related to our paper in the following part.
>
> **Q1: Different setting**
>
> A1: In our main paper, following [1] we chose a basic Transformer model to validate our conclusions, aiming to ensure the simplicity of our experiments. [1] demonstrates that the existence of scaling laws does not rely on model architecture or dataset setup. In response to the reviewer’s comments, we further validated the scaling laws on the latest DiT architecture. As demonstrated in **Supplement Material Section G**, we tested our findings on PixArt (**Supp Section G. 2**), and Flux (**Supp Section G. 2**), and also explored the impact of different resolutions (**Supp Section G. 3**) and captions (**Supp Section G. 6**).  Across various settings, our experiments show that the existence of scaling laws is independent of the Transformer architecture design and dataset characteristics.
> This paper provides a guideline on how to compute the scaling laws and how to use them. The specific model and data will not affect the scaling laws for diffusion transformers. However, the exponent and coefficient will be affected by the model and data and thus can not be directly transferred to other datasets or model architecture.  The exact scaling laws on other datasets or models should be re-computed.
>
> **Q2: Model size**
>
> A2:
> Regarding the model size, our experiments focus on models around 1B parameters. This aligns with the majority of open-sourced diffusion models, which are typically under 10B parameters, placing our model within the same order of magnitude. The primary goal of our paper is not to train a SOTA-sized diffusion model but rather to demonstrate the existence of scaling laws for diffusion models and provide a guideline on how to compute and utilize the scaling laws.
> To address the reviewer’s concerns, we suggest drawing a parallel to the development of scaling laws in the LLM domain. Scaling laws are a developing field, and the exploration of their existence and limitations has historically been incremental. For example, the initial verification of scaling laws for LLMs was conducted with models at the 600M parameter scale [1]. Over time, this expanded to tens of billions of parameters and eventually to the current 400B or more scale [2, 3]. Notably, when the first paper on scaling laws for LLM[1] was published, the largest model, GPT-3[4], had 175B parameters, creating a size ratio of **175B/0.6B ≈ 291** between SOTA models and the scaling law benchmarks.
>
> According to most reviewers’ comments and the best of our knowledge, our work is the first work to verify the existence of scaling laws in Diffusion Transformers. Most open-sourced diffusion models today are below 10B parameters, and our 1B-scale models are much closer to SOTA in this domain compared with the first few works in LLM, falling within the same order of magnitude. Scaling law validation is a step-by-step process, and the scale of models used increases over time as the field progresses.
> To put this into perspective, training a 3B diffusion model would require approximately 1.8T data tokens and close to a month of training, while a 10B model would require 6T data tokens and several months of training. This level of resource demand far exceeds the affordance of a research project. Our work focuses on proving the existence of scaling laws and providing a guideline, not on training SOTA diffusion models. Larger-scale scaling law validation, like in the LLM field, will require further exploration in future studies.
>
> [1] Scaling Laws for Neural Language Models. Kaplan et al. https://arxiv.org/abs/2001.08361
>
> [2] GPT-4 Technical Report. OpenAI. https://arxiv.org/abs/2303.08774
>
> [3] The Llama 3 Herd of Models. Dubey et al. https://arxiv.org/abs/2407.21783
>
> [4] Language Models are Few-Shot Learner. Brown et al. https://arxiv.org/abs/2005.14165

---

> ### Author Response · Authors · 2024-11-29
>
> Dear Reviewer CVDQ,
>
> We have responded to your concerns and would be happy to clarify further if needed. Please let us know if you have any additional questions or points for discussion.
>
> Thank you for your time and feedback. Wishing you a great day!
>
> Best regards,
>
> The Authors

---

> ### Author Response · Authors · 2024-12-01
>
> Dear Reviewer CVDQ,
>
> As the rebuttal deadline approaches, we would greatly appreciate hearing from you regarding any additional concerns or feedback you may have. We are more than willing to address them and engage in further discussions during the rebuttal phase.
>
> Thank you for your time and consideration.
>
> Best regards,
>
> The Authors

---

> ### Author Response · Authors · 2024-12-02
>
> Dear reviewer CVDQ,
>
> Today is the last day for discussion. We look forward to getting feedback from you regarding our response. We would love to try our best to address your concerns and do further discussion.
>
> Best,
>
> The authors

---

### Official Review · Reviewer_ssc5 · 2024-11-02

**Soundness:** 2
**Presentation:** 3
**Contribution:** 2
**Rating:** 5
**Confidence:** 4

**Summary:**

This research investigates the scaling characteristics of diffusion transformers, focusing on a range from ($1 \times 10^{17}$) to ($6 \times 10^{18}$) Flops. The study enhances the understanding of how computational effort maps to synthesis quality.

**Strengths:**

1. Good Presentation: The figures clearly present the properties of the scaling model and the associated FLOPs, making the analysis easy to follow and understand.

2. Predictable Benchmarking: The introduction of a predictable benchmark for assessing model performance and data quality significantly reduces costs and aids in optimizing computational resources.

**Weaknesses:**

1. The largest model in the study only has 1 billion parameters. While I agree with the authors' statement that "scaling laws help us make informed decisions about resource allocation to maximize computational efficiency," a model size of 1 billion parameters is too small to verify the significance of this scaling law. We need guidance for larger model sizes, especially considering that current models, such as the Flux model, already have 8 billion parameters.

2. The discussion of reference work is not accurate. Line 097 makes claims about the masked diffusion transformers but fails to mention the first Masked Diffusion Transformer (MDT), both versions v1 and v2.

**Questions:**

Are there any properties by scaling DiT across different dimensions (like channels, layers, etc)? The paper should explore how scaling DiT in various dimensions affects its properties and performance. This could provide valuable insights into the scaling behavior and help in designing more efficient models.

---

> ### Author Response · Authors · 2024-11-27
>
> We thank the reviewer for detailed comments and feedback. We respond to your concern related to our paper in the following part.
>
> **Q1: Model size**
>
> A1:
> Regarding the model size, our experiments focus on models around 1B parameters. This aligns with the majority of open-sourced diffusion models, which are typically under 10B parameters, placing our model within the same order of magnitude. The primary goal of our paper is not to train a SOTA-sized diffusion model but rather to demonstrate the existence of scaling laws for diffusion models and provide a guideline on how to compute and utilize the scaling laws.
> To address the reviewer’s concerns, we suggest drawing a parallel to the development of scaling laws in the LLM domain. Scaling laws are a developing field, and the exploration of their existence and limitations has historically been incremental. For example, the initial verification of scaling laws for LLMs was conducted with models at the 600M parameter scale [1]. Over time, this expanded to tens of billions of parameters and eventually to the current 400B or more scale [2, 3]. Notably, when the first paper on scaling laws for LLM[1] was published, the largest model, GPT-3[4], had 175B parameters, creating a size ratio of **175B/0.6B ≈ 291** between SOTA models and the scaling law benchmarks.
>
> According to most reviewers’ comments and the best of our knowledge, our work is the first work to verify the existence of scaling laws in Diffusion Transformers. Most open-sourced diffusion models today are below 10B parameters, and our 1B-scale models are much closer to SOTA in this domain compared with the first few works in LLM, falling within the same order of magnitude. Scaling law validation is a step-by-step process, and the scale of models used increases over time as the field progresses.
> To put this into perspective, training a 3B diffusion model would require approximately 1.8T data tokens and close to a month of training, while a 10B model would require 6T data tokens and several months of training. This level of resource demand far exceeds the affordance of a research project. Our work focuses on proving the existence of scaling laws and providing a guideline, not on training SOTA diffusion models. Larger-scale scaling law validation, like in the LLM field, will require further exploration in future studies.
>
> **Q2: Inaccurate reference work.**
>
> A2: We thank the reviewer for pointing out the inaccuracy of our reference work. We have modified the paper and fixed the discussion on masked diffusion models. (**MDT v1 & v2**).  The discussion can be found in **Supplementary Material Section G. 4**. We will merge the discussion into the related work part later.
>
> **Q3: Scaling Properties.**
>
> A3: The primary scope of this paper is to
> demonstrate the existence of scaling laws in diffusion transformers
> outline how these laws are computed
> how we can use these laws as a benchmark to evaluate the model/data design.
> Instead of providing the scaling properties for a specific model architecture, we provide a general method for users of scaling laws to find the scaling properties of their models. The scaling properties are normally coupled with specific model designs and should be analyzed case-by-case. Exhausting all of them are computationally intensive and beyond our scope. We leave it to future work.
>
> [1] Scaling Laws for Neural Language Models. Kaplan et al. https://arxiv.org/abs/2001.08361
>
> [2] GPT-4 Technical Report. OpenAI. https://arxiv.org/abs/2303.08774
>
> [3] The Llama 3 Herd of Models. Dubey et al. https://arxiv.org/abs/2407.21783
>
> [4] Language Models are Few-Shot Learner. Brown et al. https://arxiv.org/abs/2005.14165

---

> ### Author Response · Authors · 2024-11-29
>
> Dear Reviewer ssc5,
>
> We have responded to your concerns and would be happy to clarify further if needed. Please let us know if you have any additional questions or points for discussion.
>
> Thank you for your time and feedback. Wishing you a great day!
>
> Best regards,
>
> The Authors

---

> ### Author Response · Authors · 2024-12-01
>
> Dear Reviewer ssc5,
>
> As the rebuttal deadline approaches, we would greatly appreciate hearing from you regarding any additional concerns or feedback you may have. We are more than willing to address them and engage in further discussions during the rebuttal phase.
>
> Thank you for your time and consideration.
>
> Best regards,
>
> The Authors

---

> ### Author Response · Authors · 2024-12-02
>
> Dear reviewer ssc5,
>
> Today is the last day for discussion. We look forward to getting feedback from you regarding our response. We would love to try our best to address your concerns and do further discussion.
>
> Best,
>
> The authors

---

### Official Review · Reviewer_2p8z · 2024-11-03

**Soundness:** 3
**Presentation:** 2
**Contribution:** 3
**Rating:** 5
**Confidence:** 5

**Summary:**

This paper explore the scaling laws of DiT (for text-to-image generation). The authors experiment with different computation budget and find the optimal allocation (i.e., the lowest training loss) of model parameters and training data amount for each budget. Subsequently, they plot two relationship curves: 1) parameter-FLOPs and 2) token (seen in training)-FLOPs. The conclusion is the training loss follows a power-law relationship with the computation budget. Based on this, the author claim that they can predict the best configurations of model parameter and training data for a given (larger) budget.

**Strengths:**

1. While some similar reports have been seen in LLMs, to my best knowledge this is the first analysis for text-to-image DiTs. The problem this paper aims to address is also a hot topic troubling the community. A good question is proposed in this paper.
2. If the conclusion is correct, then given the model size and training data, it is possible to predict in which training iteration the model will achieve the best performance. This would alleviate a lot of resource consumption in the heavy task of DiT training.
3. The experimental methods are well designed.

**Weaknesses:**

1. I don't know why the authors use FLOPs and token numbers as the indicators. This makes it difficult for me to intuitively estimate the computation budgets and data. In practice, GPU hours and image amount are more common and intuitive.
2. I'm concerned about the power-law relationship shown in Fig 1 and 3, though the linear is so clear. How can we determine that this is not by coincidence (due to the structure of the model as well as the training data)? Even if this law is objectively universal, At what model size will it fail? For DiT, 1B is still a small scale. Can I really estimate 5B/10B if I only evaluate two points with small model sizes and get the regression line?
3. The training loss is not equal to FID, and more different from general performance. Of course the model does not perform well at large loss values, but it is agnostic at small values. In addition, when the budget is fixed, the number of iterations for the large model decreases. What happened if a larger learning rate is used in this case? The current setting does not seem fair.
4. The authors should improve the writing for easy understanding. For example, I can guess that the tokens in Fig. 1 refer to the number of tokens DiT has processed during the training, it still confuses me about the figure (even though the concept is explained in Sec 3.2)

**Questions:**

Please also check the concerns in Weaknesses.
1. As mentioned in W.1, why not use GPU hours and text-image pairs?
2. How can we use it in practice? Can we only estimate two points at small scale and predict the configuration at large scale? I have no confidence in it.
3. Can this conclusion be validated based on the present DiT works? They usually have the model versions with different parameters.

---

> ### Author Response · Authors · 2024-11-27
> **Response to reviewer 2p8z**
>
> We thank the reviewer for detailed comments and feedback. We respond to your concern related to our paper in the following part.
>
> **Q1: The use of FLOPs and tokens.**
>
> A1:
> We appreciate the reviewer’s feedback and understand the preference for GPU hours and image counts as intuitive indicators for computing budgets and data. While these metrics are indeed related to FLOPs and token counts, there are several reasons why we chose to use the latter:
> 1. **Consistency with Previous Work:** In prior studies on scaling laws for large language models (LLMs)[1][2][3], FLOPs and token counts are commonly used as standard metrics. By adopting these measures, we ensure consistency with the established literature and enable a direct comparison with related works.
>
> 2. **Practical Limitations of GPU Hours as a Metric:** While GPU hours can provide a sense of the computational effort, they are not a precise indicator of training capacity due to their strong dependency on the specifics of the infrastructure. GPU hours depend on factors such as the type of GPUs used, the optimization level of the compute cluster, and the utilization efficiency. For example, to translate FLOPs into GPU hours accurately, one would need to account for the **Model FLOPs Utilization (MFU)**, which reflects how efficiently GPUs are used in practice. Since MFU is highly dependent on engineering implementations and often varies across systems, GPU hours alone cannot universally represent the compute budget in a reproducible way.
>
> 3. **Tokens as a Generalized Data Representation:** Tokens provide a universal representation of the data processed by transformer-based models, as all input modalities (text, visual content, audio) are eventually tokenized. Therefore, token count serves as a more comprehensive and consistent metric for scaling compute analysis.
>
> 4. **Tokens are not equal to image count.** While image count may be a straightforward metric for fixed-resolution image datasets, it does not fully capture the scale of training data in many scenarios. In our case, we also include text tokens, which contribute significantly to the total compute cost during training. Moreover, in industrial-scale text-to-image models[4][5], training typically involves a mix of images at different resolutions. For video generation models, videos of varying lengths. These variations make it difficult to directly equate image or video counts to the total data processed. Our paper serves for a broad range of diffusion tasks including videos and images. Therefore tokens are more suitable to represent the amount of compute.
>
> **Q2: Validation on modern DiT architecture.**
>
> A2:
> First, based on prior research on large language models (LLMs)[1], the existence of scaling laws for transformer-based models is generally independent of the specific architectural design. While the model architecture may influence the coefficients and exponents of the scaling laws, it does not affect their existence.
> To address the reviewer’s concern, we conducted a series of experiments to validate our conclusions. In the main text, we have already demonstrated this through experiments on PixArt [6], and we further provide predictions of the loss in **Supplementary Material Section G.2**. Additionally, we extended our validation to the latest DiT architecture by evaluating its performance under the Flux budget. The results are included in **Supplementary Material Section G.1**. In both cases, as shown in the figures provided, there is a clear linear relationship between the compute budget and the loss, which strongly supports the presence of scaling laws.

---

> > ### Author Response · Authors · 2024-11-27
> >
> > **Q3: Concerns about the power law. When will scaling laws fail? Can we use two points for an estimation?**
> >
> > A3: The existence of the power-law scaling laws presented in our paper is not coincidental. This power-law relationship has been extensively explored and validated by several large-scale models, including GPT-4 [10], LLAMA 3.1 [1], DeepSeek [7], and Chinchilla [3], all of which observed a consistent power-law relationship between compute and loss.
> > Additionally, we conducted further experiments on various model architectures (FLUX, PixArt), resolutions (**Supplementary Section G.3**.), and datasets(**Supplementary Section G.6**.). These experiments collectively reaffirm that the scaling laws are not coincidental.
> > Scaling laws remain an active area of research, and predicting when they may no longer apply is challenging. In the realm of large language models (LLMs), scaling laws have been validated for models with up to 400 billion parameters. However, it remains uncertain whether these laws hold true for even larger models. In this paper, we extend this investigation to Diffusion Transformers, demonstrating the existence of scaling laws within this framework. Verifying whether these laws persist at larger scales, however, requires empirical validation through training models of the relevant size—a task that is currently beyond our resource capacity.
> > While two points can define a line, fitting a line with only two points is prone to high variance and reduced accuracy. It is insufficient to eliminate variance and provide an accurate representation of the underlying relationship. In practice, five or more data points are typically used to fit a line for a more reliable prediction.
> >
> > **Q4: The effect of learning rate**
> > A4: In this study, we chose a fixed learning rate of 1e−4 for the following reasons:
> > 1. **Minimal Impact of Learning Rate Variations:** As shown in Figure 2 of [7], for Transformers, the final loss remains relatively stable when the learning rate is within a reasonable range. Also, empirically, 1e-4 is typically used in previous DiT models[13] and training scripts in repos such as diffusers as the default learning rate.  Previous work has found 1e-4 to be generally good across several model sizes. To further demonstrate this, we train the largest model in our main paper under various learning rates. The results, summarized in the **Supplementary Material G.5**.It show that 1e-4 consistently performs well and within the optimal range. Increasing or decreasing the learning rate beyond this range, however, leads to poorer performance.
> >
> > 2. **Instability of Larger Models with Higher Learning Rates:** Larger Transformers are known to be highly sensitive to learning rate settings. As discussed in [8][9], stabilizing the training of large-scale Transformers often requires lower learning rates and techniques like learning rate warmup to synchronize updates across deeper and shallower layers. Using a higher learning rate, especially in the early stages of training, can lead to instability and degraded model performance.
> >
> > We agree with the reviewer's opinion that the learning rate may affect the results. However, as long as the learning rate is neither too large nor too small, it will only cause some fluctuations and will not impact the existence of scaling laws.
> >
> > Q5: **Practical use case**
> >
> > A5: Scaling laws can assist in predicting optimal resource allocations and the trained model’s loss and FID. To address the reviewer’s concern, we provide a practical use case. Typically, companies determine the total training time for their final models. As mentioned earlier, once the model architecture and training infrastructure are fixed, the Model FLOPs Utilization (MFU) can be calculated, This reflects the efficiency of GPU utilization. Given the type of GPU, we can compute the total floating-point operations achievable within the specified time, representing the total compute budget.
> > Using this budget, we define a range of smaller compute values (e.g., 10^{17} to 10^{19}) and train models of varying sizes under each budget, as outlined in **Section 3.2** of our paper. By following this procedure, we can derive scaling curves for metrics such as loss, parameters, tokens, and FID. From these fitted scaling curves, we can predict the optimal allocation of model size and data quantity, identifying the best parameters and dataset size for the final model. Additionally, the scaling curves enable us to estimate the expected loss or FID for even larger models, providing valuable foresight for resource planning.

---

> ### Author Response · Authors · 2024-11-27
>
> Q5: **Model size**
>
> A5:
> Regarding the model size, our experiments focus on models around 1B parameters. This aligns with the majority of open-sourced diffusion models, which are typically under 10B parameters, placing our model within the same order of magnitude. The primary goal of our paper is not to train a SOTA-sized diffusion model but rather to demonstrate the existence of scaling laws for diffusion models and provide a guideline on how to compute and utilize the scaling laws.
>
> To address the reviewer’s concerns, we suggest drawing a parallel to the development of scaling laws in the LLM domain. Scaling laws are a developing field, and the exploration of their existence and limitations has historically been incremental. For example, the initial verification of scaling laws for LLMs was conducted with models at the 600M parameter scale [1]. Over time, this expanded to tens of billions of parameters and eventually to the current 400B or more scale [10, 11]. Notably, when the first paper on scaling laws for LLM[1] was published, the largest model, GPT-3[12], had 175B parameters, creating a size ratio of **175B/0.6B ≈ 291** between SOTA models and the scaling law benchmarks.
>
> According to most reviewers’ comments and the best of our knowledge, our work is the first work to verify the existence of scaling laws in Diffusion Transformers. Most open-sourced diffusion models today are below 10B parameters, and our 1B-scale models are much closer to SOTA in this domain compared with the first few works in LLM, falling within the same order of magnitude. Scaling law validation is a step-by-step process, and the scale of models used increases over time as the field progresses.
>
> Training a 3B diffusion model would require approximately 1.8T data tokens and close to a month of training, while a 10B model would require 6T data tokens and several months of training. This level of resource demand far exceeds the affordance of a research project. Our work focuses on proving the existence of scaling laws and providing a guideline, not on training SOTA diffusion models. Larger-scale scaling law validation, like in the LLM field, will require further exploration in future studies.
>
> Q7: **Training loss and FID?**
>
> A7: Training loss and FID are indeed different. They focus on different aspects of evaluations but are strongly correlated. In our paper, we demonstrate the existence of scaling laws both for loss and FID. We can predict the large model’s FID using small-scale models.
> Training loss serves as both a reliable predictor and an evaluation metric. As demonstrated in **SD3 [4] Figure 8**, validation loss strongly correlates with overall model performance, showing a significant relationship between loss values and holistic image evaluation metrics. In our study, **Figure 2** illustrates that training loss and validation loss exhibit similar trends and shapes, confirming that scaling laws can accurately predict validation loss. Consequently, training or validation loss can be used as an indicator of a model's quality.
> Scaling laws also provide guidance for identifying the optimal model size, allowing us to determine the appropriate parameter range. However, model parameters are typically discrete, making it impractical to precisely match the optimal size suggested by the scaling laws. From Figure 1, we observe that loss does not change significantly in the vicinity of the optimal point, ensuring that selecting a size close to the predicted optimal still yields near-optimal performance. However, for diffusion models, even a small difference in loss, such as 0.05, can result in substantial performance variations. Therefore, if the chosen model size deviates too far from the optimal, the seemingly minor numerical loss differences can translate into significant practical performance disparities.
>
> Q8: **Writing quality.**
>
> A8: We thank the reviewer for pointing out the writing issues in our paper. We will improve the writing to make the paper more clear.

---

> > ### Author Response · Authors · 2024-11-27
> >
> > [1] Scaling Laws for Neural Language Models. Kaplan et al. https://arxiv.org/abs/2001.08361
> >
> > [2] Scaling Laws for Autoregressive Generative Modeling. Henighan et al. https://arxiv.org/abs/2010.14701
> >
> > [3] Training Compute-Optimal Large Language Models. Hoffmann et al. https://arxiv.org/abs/2203.15556
> >
> > [4] Scaling Rectified Flow Transformers for High-Resolution Image Synthesis. Esser et al. https://arxiv.org/abs/2403.03206
> >
> > [5] SDXL: Improving Latent Diffusion Models for High-Resolution Image Synthesis. Podell et al. https://arxiv.org/abs/2307.01952
> >
> > [6] PixArt-α: Fast Training of Diffusion Transformer for Photorealistic Text-to-Image Synthesis. Chen et al. https://arxiv.org/abs/2310.00426
> >
> > [7] DeepSeek LLM: Scaling Open-Source Language Models with Longtermism. Bi et al. https://arxiv.org/abs/2401.02954
> >
> > [8] Small-scale proxies for large-scale Transformer training instabilities. Wortsman et al. https://arxiv.org/abs/2309.14322
> >
> > [9] DeepNet: Scaling Transformers to 1,000 Layers. Wang et al. https://arxiv.org/abs/2203.00555
> >
> > [10] GPT-4 Technical Report. OpenAI. https://arxiv.org/abs/2303.08774
> >
> > [11] The Llama 3 Herd of Models. Dubey et al. https://arxiv.org/abs/2407.21783
> >
> > [12] Language Models are Few-Shot Learner. Brown et al. https://arxiv.org/abs/2005.14165
> >
> > [13] Scalable Diffusion Models with Transformers. Peebles et al. https://arxiv.org/abs/2212.09748

---

> ### Author Response · Authors · 2024-11-29
>
> Dear Reviewer 2p8z,
>
> We have responded to your concerns and would be happy to clarify further if needed. Please let us know if you have any additional questions or points for discussion.
>
> Thank you for your time and feedback. Wishing you a great day!
>
> Best regards,
>
> The Authors

---

> > ### Comment · Reviewer_2p8z · 2024-11-29
> > **Response to authors**
> >
> > For Q1, maybe I didn't express myself clearly enough. I agree with using FLOPs and token numbers as standard metrics, but it would be better to provide gpu hours and image numbers as reference. Otherwise very few readers will be able to quickly understand those numbers.
> >
> > Considering other reviewer's comments and the author's responses, I think that the current experiments cannot strongly support the conclusions so i would like to keep my rating.

---

> > > ### Author Response · Authors · 2024-11-30
> > > **Response to reviewer 2p8z**
> > >
> > > Dear Reviewer 2p8z,
> > >
> > > Thank you for your valuable feedback. We sincerely appreciate your comments and suggestions, which we believe will greatly help in improving our work.
> > >
> > > Regarding Q1, we completely agree that adding additional notation may enhance the clarity of our explanations. Thank you for bringing this to our attention.
> > >
> > > Could you kindly elaborate further on your comment that **current experiments cannot strongly support the conclusions**? We would like to better understand your concerns so we can address them effectively.
> > >
> > > For your concern about **whether this is a coincidence**, we conducted experiments across **three architectures and several data settings**, all of which consistently demonstrated a clear linear trend. Based on these results, we believe this strongly indicates that the observed phenomenon **is not a coincidence**. We would kindly suggest the reviewer read prior works on scaling laws in LLMs [1][2][3], which show that such patterns exist **independently of coincidence**.
> > >
> > > Regarding the model size, our largest model is of the same order of magnitude as current SOTA models. As explained in our paper, there is no strict need to train a full-scale SOTA model to establish the scaling laws, and we drew a parallel with similar observations in LLM scaling laws. So in previous LLM, the **model ratio size in LLM is 291** which is much larger than our paper, and the conclusion in LLM holds even on the current largest models.
> > >
> > > On the topic of two-point estimations, we recommend consulting prior works on scaling laws. Using only two points can lead to **high-variance estimations**, making the results unreliable and impractical for robust conclusions. Hence, a broader evaluation framework, as used in our study, is more appropriate and ensures greater reliability.
> > >
> > > Finally, we would like to better understand your perspective on strengthening the experimental support for our conclusions. Could you kindly provide more specific guidance on what additional experiments or analyses you believe would help address your concerns (e.g., larger model sizes, further verification against coincidence, or other considerations)?
> > >
> > > We sincerely look forward to your feedback and are committed to addressing your concerns in the most constructive way possible.
> > > Best regards, The Authors
> > >
> > > [1] Scaling Laws for Neural Language Models. Kaplan et al. https://arxiv.org/abs/2001.08361
> > >
> > > [2] Scaling Laws for Autoregressive Generative Modeling. Henighan et al. https://arxiv.org/abs/2010.14701
> > >
> > > [3] Training Compute-Optimal Large Language Models. Hoffmann et al. https://arxiv.org/abs/2203.15556

---

> ### Author Response · Authors · 2024-12-01
>
> Dear Reviewer 2p8z,
>
> As the rebuttal deadline approaches, we would greatly appreciate hearing from you regarding any additional concerns or feedback you may have. We are more than willing to address them and engage in further discussions during the rebuttal phase. Besides, we have some more discussions with other reviewers, and perhaps some of them can also help solve your concerns.
>
> Thank you for your time and consideration.
>
> Best regards,
>
> The Authors

---

> ### Author Response · Authors · 2024-12-02
>
> Dear reviewer 2p8z,
>
> Today is the last day for discussion. We look forward to getting feedback from you regarding our response. We would love to try our best to address your concerns and do further discussion.
>
> Best,
>
> The authors

---

### Author Response · Authors · 2024-11-27
**Response to common concerns**

We thank all reviewers for providing constructive advice. There are some common concerns raised by multiple reviewers. We respond to the common concerns here.

Q1: **Model size**

A1:
Regarding the model size, our experiments focus on models around 1B parameters. This aligns with the majority of open-sourced diffusion models, which are typically under 10B parameters, placing our model within the same order of magnitude. The primary goal of our paper is not to train a SOTA-sized diffusion model but rather to demonstrate the existence of scaling laws for diffusion models and provide a guideline on how to compute and utilize the scaling laws.
To address the reviewer’s concerns, we suggest drawing a parallel to the development of scaling laws in the LLM domain. Scaling laws are a developing field, and the exploration of their existence and limitations has historically been incremental. For example, the initial verification of scaling laws for LLMs was conducted with models at the 600M parameter scale [1]. Over time, this expanded to tens of billions of parameters and eventually to the current 400B or more scale [2, 3]. Notably, when the first paper on scaling laws for LLM[1] was published, the largest model, GPT-3[4], had 175B parameters, creating a size ratio of **175B/0.6B ≈ 291** between SOTA models and the scaling law benchmarks.

According to most reviewers’ comments and the best of our knowledge, our work is the first work to verify the existence of scaling laws in Diffusion Transformers. Most open-sourced diffusion models today are below 10B parameters, and our 1B-scale models are much closer to SOTA in this domain compared with the first few works in LLM, falling within the same order of magnitude. Scaling law validation is a step-by-step process, and the scale of models used increases over time as the field progresses.
To put this into perspective, training a 3B diffusion model would require approximately 1.8T data tokens and close to a month of training, while a 10B model would require 6T data tokens and several months of training. This level of resource demand far exceeds the affordance of a research project. Our work focuses on proving the existence of scaling laws and providing a guideline, not on training SOTA diffusion models. Larger-scale scaling law validation, like in the LLM field, will require further exploration in future studies.

Q2: **Model architecture.**

A2: First, based on prior research on large language models (LLMs)[1](Figure 5.), the existence of scaling laws for transformer-based models is generally independent of the specific architectural design. While the model architecture may influence the coefficients and exponents of the scaling laws, it does not affect their existence.
To address the reviewer’s concern, we conducted a series of experiments to validate our conclusions. In the main text, we have already demonstrated this through experiments on PixArt [5], and we further provide predictions of the loss in **Supplementary Section G.2**. Additionally, we extended our validation to the latest DiT architecture by evaluating its performance under the Flux budget. The results are included in **Supplementary Section G.1** as well. In both cases, as shown in the figures provided, there is a clear linear relationship between the compute budget and the loss, which strongly supports the presence of scaling laws.

[1] Scaling Laws for Neural Language Models. Kaplan et al. https://arxiv.org/abs/2001.08361

[2] GPT-4 Technical Report. OpenAI. https://arxiv.org/abs/2303.08774

[3] The Llama 3 Herd of Models. Dubey et al. https://arxiv.org/abs/2407.21783

[4] Language Models are Few-Shot Learner. Brown et al. https://arxiv.org/abs/2005.14165

[5] PixArt-α: Fast Training of Diffusion Transformer for Photorealistic Text-to-Image Synthesis. Chen et al. https://arxiv.org/abs/2310.00426

---

### Author Response · Authors · 2024-12-04
**General Response**

We sincerely thank the reviewers for carefully reading our paper and providing insightful comments. We are grateful for the recognition of our work as “the first scaling laws for DiT” (Reviewer 2p8z, CVDQ) and for its practical value in “DiT training” (Reviewer 2p8z, ssc5, CVDQ, CvMZ). The acknowledgment of our contributions to providing a “predictable benchmark” (Reviewer ssc5, CvMz), “alleviating resource consumption,” and employing “well-designed methods” (Reviewer 2p8z) is highly appreciated. Additionally, we are pleased that the presentation quality has been described as “good” (Reviewer ssc5).
In response to the reviewers’ feedback, we have made modifications to our manuscript, marking all revisions in blue for clarity. These updates include

* To address the reviewers’ concern about whether the scaling laws apply to modern DiT, we conducted experiments on modern model architecture FLUX and Pixart. (Supp Section G.1 and G.2. Reviewer 2p8z, CvMz).

* To demonstrate that the learning rate will not affect the existence of scaling laws and that a larger learning rate for models with more parameters is not optimal, we conduct a learning rate sensitivity analysis. (Supp Section G.5, reviewer 2p8z.)

* To complete the predictable benchmark, we add experiments to show how we can asses data using scaling laws (Supp Section G.6. Reviewer CvMz)

* To eliminate the reviewer’s concern about the resolution dependency, we conduct experiments on 512 resolution. (Supp Section G.3. Reviewer CvMz)

* We added a more accurate discussion on related work of masked diffusion transformers. (Supp Section G.4. Reviewer ssc5)

* Reviewers’ main concern of the paper lies in the model size, we have responded in the common concerns part.

We hope our response can help address the reviewers’ concerns raised before and during the discussion.

---

### Meta-Review · Area_Chair_Qh5z · 2024-12-05

**Metareview:**

### **Summary**
This paper investigates scaling laws for Diffusion Transformers (DiT) in the context of text-to-image generation. The authors conduct experiments to reveal a power-law relationship between training loss and compute budget and claim these findings can inform optimal model sizes and data requirements for future research. Additionally, the authors demonstrate that scaling laws can serve as benchmarks for evaluating data and model architectures.

While the topic is highly relevant to the community, the submission falls short of acceptance standards due to the following major concerns:


### **Strengths**
1. **Novelty**:  This is one of the first studies to explore scaling laws for DiT, which has potential implications for future research and resource optimization.
2. **Relevance**:  Scaling laws are an important topic in machine learning, and applying these insights to diffusion models aligns well with the current state of research.
3. **Clarity of Presentation**:   Visualizations and explanations of scaling relationships are clear and well-organized.
4. **Practical Implications**:   The idea of using scaling laws to assess data and model quality is valuable.


### **Weaknesses**
1. **Limited Experimental Scope**:
   - The experiments are conducted with models capped at 1B parameters, far smaller than the typical scale for state-of-the-art diffusion models (~10B). This raises concerns about the generalizability of the findings.
   - The compute budget range (spanning three orders of magnitude) is narrow compared to existing works in the domain, which span seven or more orders of magnitude.

2. **Insufficient Baseline Comparisons**:
   - There is no comparison with state-of-the-art diffusion models, such as those using more modern architectures (e.g., FLUX or SD3).
   - The analysis does not provide meaningful insights into how these scaling laws hold under diverse architectural or data settings, limiting the work’s relevance to current practical setups.

3. **Questionable Use of Metrics**:
   - While FLOPs and tokens are reasonable metrics in theoretical studies, they lack practical interpretability. The absence of GPU hours or image counts as supplementary references makes the findings less accessible to practitioners.
   - The connection between training loss and downstream generation metrics (e.g., FID) is weakly substantiated, raising concerns about the reliability of conclusions drawn from loss trends alone.


### **Recommendation**
The work is novel and timely, but the experimental design and scope are insufficient to support the authors’ claims. The findings are limited in their generalizability and applicability to modern diffusion models, and key methodological concerns remain unaddressed. While this paper provides an initial exploration of scaling laws for diffusion transformers, it does not quite address the concerns of reviewers in its current form.

**Additional Comments On Reviewer Discussion:**

During the rebuttal, reviewers raised concerns about the limited scope, including the use of small models (≤1B parameters) and a narrow compute budget range. While the authors justified these constraints by drawing parallels to early scaling law studies, the limited scale reduces the broader applicability of the findings. Reviewers also questioned the use of FLOPs and tokens as metrics, suggesting the inclusion of more practical references like GPU hours and image counts.

The generality of the scaling laws across modern architectures was challenged. In response, the authors added experiments with architectures like FLUX and PixArt but acknowledged the need to recompute coefficients for different settings. Concerns about misleading claims comparing cross-attention and in-context learning were partially addressed through clarifications.

Despite these efforts, the reviewers remained unconvinced about the robustness of the connection between loss and downstream metrics like FID and the study’s practical relevance. While the paper provides an interesting starting point, its limited experimental scope, lack of practical utility, and methodological shortcomings prevent it from meeting the acceptance threshold. **Decision: Reject.**

---

### Decision · Program_Chairs · 2025-01-22

Reject